# Bile acid 7α-dehydroxylating bacteria accelerate injury-induced mucosal healing in the colon

Antoine Jalil[1,5], Alessia Perino [ID][1,5], Yuan Dong [ID][2], Jéromine Imbach[1], Colin Volet [ID][3], Eduard Vico-Oton[3], Hadrien Demagny[1], Lucie Plantade[1], Hector Gallart-Ayala[4], Julijana Ivanisevic[4], Rizlan Bernier-Latmani [ID][3], Siegfried Hapfelmeier [ID][2] & Kristina Schoonjans [ID][1✉]

## Abstract

Host-microbiome communication is frequently perturbed in gut pathologies due to microbiome dysbiosis, leading to altered production of bacterial metabolites. Among these, 7α-dehydroxylated bile acids are notably diminished in inflammatory bowel disease patients. Herein, we investigated whether restoration of 7α-dehydroxylated bile acids levels by *Clostridium scindens*, a human-derived 7α-dehydroxylating bacterium, can reestablish intestinal epithelium homeostasis following colon injury. Gnotobiotic and conventional mice were subjected to chemically-induced experimental colitis following administration of *Clostridium scindens*. Colonization enhanced the production of 7α-dehydroxylated bile acids and conferred prophylactic and therapeutic protection against colon injury through epithelial regeneration and specification. Computational analysis of human datasets confirmed defects in intestinal cell renewal and differentiation in ulcerative colitis patients while expression of genes involved in those pathways showed a robust positive correlation with 7α-dehydroxylated bile acid levels. *Clostridium scindens* administration could therefore be a promising biotherapeutic strategy to foster mucosal healing following colon injury by restoring bile acid homeostasis.

**Keywords** Bile Acids; 7α-dehydroxylating Bacteria; Intestinal Mucosal Healing; Ulcerative Colitis

**Subject Categories** Digestive System; Microbiology, Virology & Host Pathogen Interaction

## Introduction

The intricate interplay between the gut microbiome and the host is critical for maintaining overall whole-body homeostasis. Disruption of these finely tuned interactions can lead to a plethora of pathologies, including gastrointestinal diseases. Among these, inflammatory bowel diseases (IBD) are chronic and debilitating gastrointestinal disorders, and their prevalence is rising exponentially, constituting a substantial societal burden (Alatab et al, 2020). IBD, defined by chronic relapsing inflammation of the gastrointestinal tract, includes specific subtypes like Crohn's disease (CD) and ulcerative colitis (UC) (Chang, 2020). While the exact etiology of IBD remains incompletely understood, prevailing theories underscore the significant impact of environmental factors on genetically predisposed individuals (Chang, 2020; Graham and Xavier, 2020). Within this framework, the intestinal microbiome emerges as a key actor. Dysbiosis, characterized by a significant change in the microbial composition and associated alterations in bacterial metabolites, is increasingly recognized as a key characteristic of IBD (Lavelle and Sokol, 2020).

The gut microbial community produces numerous metabolites that play crucial roles in regulating host physiology. Out of this large pool, secondary bile acids (BAs) have emerged as prominent signaling molecules (Krautkramer et al, 2021). BAs are initially synthesized in the liver as primary BAs and released into the intestine after a meal to solubilize dietary lipids (Perino et al, 2020). Primary BAs are efficiently reabsorbed in the distal part of the small intestine and recycled to the liver, yet small amounts transit to the colon where they are converted into secondary BAs by resident gut bacteria, through multiple enzymatic modifications (Collins et al, 2022; Guzior and Quinn, 2021; Wahlström et al, 2016). Bacteria with 7α-dehydroxylating activity, such as *Clostridium scindens* (*C. scindens*), typically transform unconjugated primary BAs, cholic acid (CA) and chenodeoxycholic acid (CDCA), into the major bacterial-derived secondary BAs, deoxycholic acid (DCA) and lithocholic acid (LCA), respectively (Ridlon et al, 2023).

Previous studies have shown that dysbiotic UC patients have lower levels of fecal 7α-dehydroxylated BAs compared to healthy individuals (Sinha et al, 2020; Duboc et al, 2013; Lloyd-Price et al, 2019), suggesting that local BA conversion in the colon could be compromised. The signaling function of 7α-dehydroxylated BAs relies on dedicated receptors (Perino and Schoonjans, 2022), such as Takeda-G-protein-receptor-5 (TGR5, GPBAR1) (Kawamata et al, 2003), whose activation drives host processes, including gut hormone secretion (Thomas et al, 2009; Kuhre et al, 2018;

[1]Laboratory of Metabolic Signaling, Institute of Bioengineering, School of Life Sciences, Ecole Polytechnique Fédérale de Lausanne, Lausanne, Switzerland. [2]Institute for Infectious Diseases, University of Bern, Bern, Switzerland. [3]Environmental Microbiology Laboratory, School of Architecture, Civil and Environmental Engineering, Ecole Polytechnique Fédérale de Lausanne, Lausanne, Switzerland. [4]Metabolomics Platform, Faculty of Biology and Medicine, University of Lausanne, Lausanne, Switzerland. [5]These authors contributed equally: Antoine Jalil, Alessia Perino. ✉E-mail: kristina.schoonjans@epfl.ch

Brighton et al, 2015), immunomodulation (Sinha et al, 2020; Biagioli et al, 2017; Cipriani et al, 2011; Garibay et al, 2019), and stem cell-induced intestinal renewal (Sorrentino et al, 2020). Building on these premises, we sought to investigate whether administration of *C. scindens* could increase 7α-dehydroxylation capacity of the gut and accelerate mucosal healing in models of experimental colitis, and to explore the relevance of this mechanism in human UC.

# Results

## *C. scindens* colonization increases 7α-dehydroxylated BA production and diminishes DSS-induced epithelial injury in gnotobiotic Oligo-MM¹² mice

7α-dehydroxylated BAs (e.g., DCA and LCA) are generated by a highly restricted taxonomic group of bacteria, exemplified by the human-derived bacterium *C. scindens* (Mallonee et al, 1990; Marion et al, 2019; Ridlon and Hylemon, 2012; Studer et al, 2016), that harbors the *BA-inducible* operon required for catalyzing the BA 7α-dehydroxylation reaction (Figs. 1A and EV1A) (Guzior and Quinn, 2021; Ridlon et al, 2006; Funabashi et al, 2020). To explore whether the reduction in 7α-dehydroxylated BAs observed in IBD (Sinha et al, 2020; Duboc et al, 2013; Lloyd-Price et al, 2019) could be explained by an alteration in the abundance of 7α-dehydroxylating bacteria, we generated Oligo Mouse Microbiota 12 (Oligo-MM¹²) mice, a gnotobiotic mouse model established by colonizing germ-free mice with a consortium of 12 bacterial isolates representing major phyla of the murine intestinal microbiota, but devoid of microbial 7α-dehydroxylation activity (Brugiroux et al, 2016). We then colonized this model with *C. scindens* ATCC 35704 (Oligo-MM¹² + *C. scindens*) (Fig. 1B,C) and compared the abundance of several BA species in the feces of the control and *C. scindens* experimental cohorts (Table EV1). Amendment of Oligo-MM¹² mice with *C. scindens* elicited the production of 7α-dehydroxylated BAs (Fig. 1D,E (left panel)), including DCA (Fig. 1E (right panel) and LCA (Fig. EV1B (bottom panel)). *C. scindens* also increased the proportion and amounts of at least 5 other DCA- or LCA-derivatives (Figs. 1D, EV1C, Table EV1). Conversely, as expected, primary BA proportion (Fig. 1F (left panel), EV1C)), CA (Fig. 1F (right panel)), and CDCA (Fig. EV1B (upper panel)) levels were decreased. Accordingly, colonization with *C. scindens* resulted in a higher secondary-to-primary BA ratio (Fig. 1G (left panel)). The total pool of fecal BAs, however, did not differ according to the presence or absence of *C. scindens* (Fig. 1G (right panel)). Similarly, colonization with *C. scindens* led to a significant increase in the proportion and abundance of 7α-dehydroxylated BAs (Fig. EV1D,E), resulting in a higher secondary-to-primary BA ratio (Fig. EV1F (left panel)), without affecting the total plasma BA pool (Fig. EV1F (right panel)). Of note, these selective changes in BA species (Table EV2) were insufficient to alter the BA hydrophobicity index (Fig. EV1G). Furthermore, hepatic or ileal transcript levels of key genes involved in the synthesis, signaling and transport of BAs were not significantly altered in mice colonized with *C. scindens* (Fig. EV1H,I). Altogether, these data indicate that the metabolic activity of this bacterium influences the BA composition but not the size of the BA pool in the fecal and plasma compartments.

To evaluate the impact of *C. scindens* colonization on mucosal healing, we induced colonic epithelial injury in Oligo-MM¹² mice by administering dextran sulfate sodium (DSS) followed by a 3-day recovery phase (Fig. 1H). Remarkably, the body weight of the Oligo-MM¹² mice colonized with *C. scindens* began to increase on the day after DSS withdrawal, in sharp contrast to that of the control mice, which continued to decrease until day 8 and remained lower throughout the DSS treatment and recovery phase (Fig. 1I). Introduction of *C. scindens* promoted intestinal regeneration, as evidenced by the increased number of EdU⁺-proliferating cells in the colon of *C. scindens*-colonized Oligo-MM¹² mice subjected to DSS (Fig. 1J,K). As expected, EdU⁺ cells were not increased in the unchallenged jejunum or ileum of *C. scindens*-colonized Oligo-MM¹² mice, yet augmented in the colon (Fig. EV1J,K). Overall, these data show that *C. scindens* colonization in Oligo-MM¹² mice is associated with significantly enhanced regenerative proliferative response during epithelial healing.

## *C. scindens* colonization alleviates DSS-induced epithelial injury in SPF mice

To extend our findings to a model with a more complex microbiome, we used C57BL/6J specific pathogen-free (SPF) conventional mice. 10-week-old male mice were either gavaged with live *C. scindens* bacteria at 3 different doses (10⁸, 10⁹ or 10¹⁰ colony-forming units (CFU)) for 15 consecutive days (Fig. 2A), or for 5 days after preconditioning with vancomycin (500 mg/L in drinking water) for 7 days (SPF-Van) (Ma et al, 2018) (Fig. 2B). We selected vancomycin for its antimicrobial spectrum against Gram⁺ bacteria (Watanakunakorn, 1984), and its capacity to trigger dysbiosis, which could facilitate *C. scindens* engraftment and colonization (Ma et al, 2018). Quantification of fecal *C. scindens* abundance in the different experimental conditions revealed that administration of 10⁸ CFU to SPF-Van mice was the optimal protocol for intestinal colonization by *C. scindens* (Fig. 2C). Of note, colonization of SPF-Van mice with *C. scindens* (10⁸ CFU) led to a significant increase in 7α-dehydroxylating activity in feces. This was evidenced by the elevated conversion of exogenous CA to DCA by fecal bacteria from *C. scindens*-colonized mice compared to controls during a 24-h in vitro assay (Fig. 2D). Once established the optimal colonization conditions, we next performed 16S rRNA gene amplicon sequencing to investigate the effects of vancomycin preconditioning and *C. scindens* administration on microbiome composition. As expected, vancomycin reduced fecal microbial community diversity (Fig. EV2A) and increased the dysbiosis index (Fig. EV2B). Non-metric multidimensional scaling analysis showed a distinct separation between SPF-Van mice treated with vehicle and those colonized with *C. scindens* (Fig. EV2C). Moreover, the bacterial family composition was modulated significantly by vancomycin preconditioning, characterized by the emergence of Lachnospiraceae, Akkermansiaceae, Rumminococcaceae and Anaeroplasmataceae families and the eradication of the Muribaculaceae family (Fig. EV2D). Of note, *C. scindens* administration did not significantly affect the overall family composition (Fig. EV2D). Recapitulating our key observations in the Oligo-MM¹² model, *C. scindens* colonization increased fecal and plasma 7α-dehydroxylated BA proportion (Figs. 2E and E-V2E,F (in red)) and amount (Fig. EV2F,G) in SPF-Van mice. In contrast, the fecal proportion (Fig. EV2E,F) and plasma

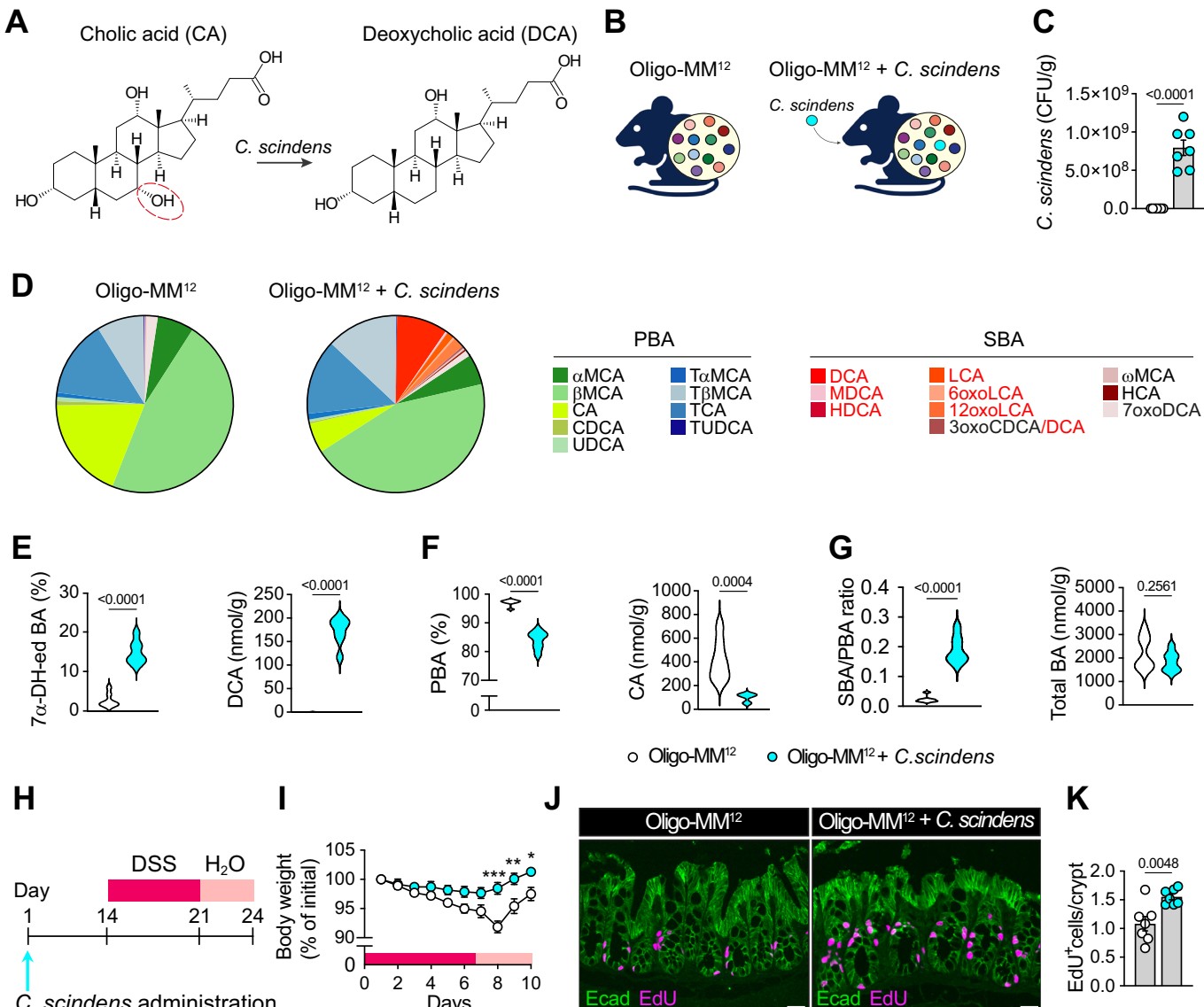

**Figure 1.  Colonization of Oligo-MM$^{12}$ mice with *C. scindens* ameliorates DSS-induced experimental colitis.**

(**A**) 7α-dehydroxylation reaction at carbon 7 (in red) of CA to form DCA mediated by *C. scindens*. (**B**) Microbiome composition of Oligo-MM$^{12}$ mice colonized or not with *C. scindens*. (**C**) *C. scindens* abundance in feces of 8-week-old male Oligo-MM$^{12}$ mice gavaged with live *C. scindens* ($10^7$ CFU – Oligo-MM$^{12}$ + *C. scindens*) or PBS (control – Oligo-MM$^{12}$) 10 days after the experiment start ($n = 7$/group) (Oligo-MM$^{12}$ vs Oligo-MM$^{12}$ + *C. scindens*: $P < 0.0001$). (**D**) Fecal BA composition of mice in (**C**). BA data are normalized to the total amount of BAs. 7α-dehydroxylated BAs are in red. (**E–G**) Amount (nmol/g) or proportion (percentage of total BA amount) of the indicated BAs and the secondary (SBA)-to-primary (PBA) BA ratio in feces of mice in (**C**). For percentage, BA data are normalized to the total amount of BAs. 7α-DH-ed: 7α-dehydroxylated BAs. (For 7α-DH-ed BA (%), DCA (nmol/g), PBA (%) and SBA/PBA ratio: Oligo-MM$^{12}$ vs Oligo-MM$^{12}$ + *C. scindens* $P < 0.0001$). (**H**) 8-week-old male Oligo-MM$^{12}$ mice were gavaged with live *C. scindens* ($10^7$ CFU) or PBS (control) 14 days before a 7-day treatment with DSS (2.75% in drinking water) followed by a 3-day recovery period without DSS in drinking water ($n = 8$/group). (**I**) Percentage of body weight loss of mice in (**H**) (Oligo-MM$^{12}$ vs Oligo-MM$^{12}$ + *C. scindens* day 8: ***$P = 0.0003$; day 9: **$P = 0.009$; day 10: *$P = 0.014$). (**J, K**) Representative images (**J**) and quantification of EdU$^+$ cells per crypt (**K**) in the colon of mice in (**H**) (day 3 after DSS withdrawal). Scale bar = 30 μm. Graphs represent mean ± SEM. $n$ refers to biological replicates. $P$ values (exact values) were calculated using two-way ANOVA followed by Bonferroni's post hoc correction (**I**) or 2-tailed Student's t-test (**C, E, F, G, K**). Source data are available online for this figure.

concentration (Fig. EV2G) of primary BAs decreased compared with control SPF-Van mice, with a consequent increase in the secondary-to-primary BA ratio (Figs. EV2H,I). The total fecal (Fig. EV2H) and plasma (Fig. EV2I) BA pool remained unaffected, with changes in selective BA species (Tables EV3, 4) that were not sufficient to affect the plasma BA hydrophobicity index (Fig. EV2J).

We then exposed SPF-Van mice to DSS to induce colonic epithelial damage and associated experimental colitis (Fig. 2F). Of note, colonization of *C. scindens* in SPF-Van mice significantly alleviated disease severity, which was reflected by reduced body weight loss (Fig. 2G), increased colon length (Fig. 2H,I) and reduced histopathologic score (Fig. 2J,K). In line with these findings, *C. scindens* colonization was associated with improved

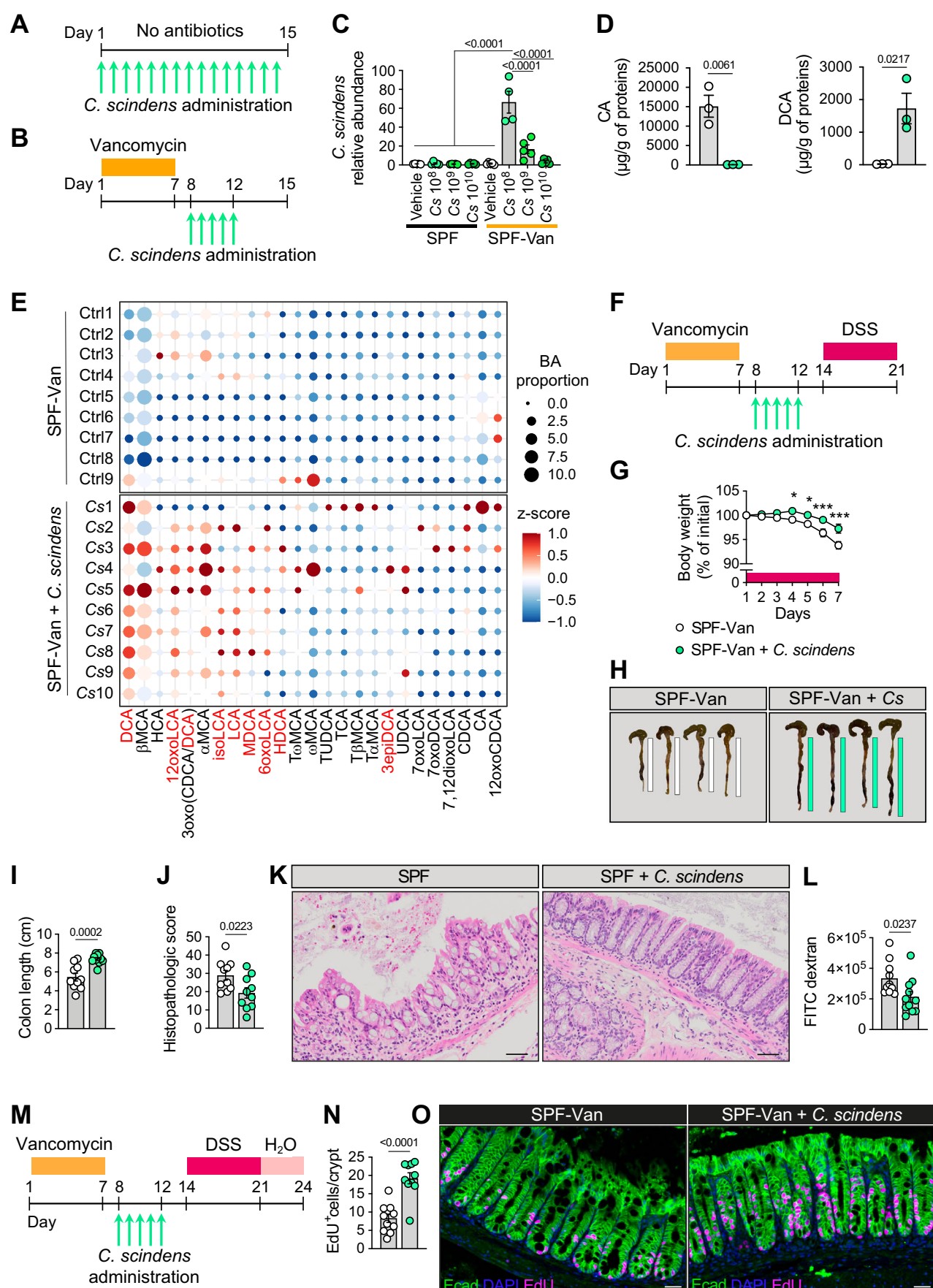

**Figure 2.  *C. scindens* administration alleviates DSS-induced experimental colitis and improves intestinal regeneration in SPF-Van mice.**

(A, B) 10-week-old C57BL/6J male mice (SPF) were gavaged daily for 15 days (**A**) or for 5 days after preconditioning with vancomycin for 7 days (SPF-Van) (**B**) with live *C. scindens* at 3 different doses ($10^8$, $10^9$ or $10^{10}$ CFU) or vehicle (PBS) ($n = 5$/group except for $n = 4$ for SPF-Van Cs $10^8$). (**C**) Quantification of the relative abundance of *C. scindens* (Cs) in feces of mice in (**A**) and (**B**) at the end of the experiment (SPF-Van Cs $10^8$ vs all other groups $P < 0.0001$). (**D**) In vitro DCA conversion from CA by fresh fecal bacteria in 24-h culture. CA and DCA amounts were normalized to fecal proteins at the end of the culture. Fresh fecal pellets were collected from SPF-Van mice colonized or not with *C. scindens* $10^8$ CFU ($n = 3$). (**E**) Proportion of each BA species over the total BA amount (dot size) and z-score (dot color) in feces at the end of the experiment. 7α-dehydroxylated BAs are in red. Ctrl = SPF-Van ($n = 9$) and Cs = SPF-Van + *C. scindens* $10^8$ CFU ($n = 10$). (**F**) 10-week-old male SPF-Van mice were gavaged daily for 5 days with live *C. scindens* ($10^8$ CFU – SPF-Van + *C. scindens*) ($n = 11$) or PBS (control – SPF-Van) ($n = 10$) and experimental colitis was induced by a 7-day treatment with DSS (2.5% in drinking water). (**G**) Percentage of body weight loss of mice in (**F**) (SPF-Van vs SPF-Van + *C. scindens* day 4: *$P = 0.019$, day 5: *$P = 0.019$, day 6: ***$P < 0.0001$, day 7: ***$P < 0.0001$). (**H, I**) Representative images (**H**) and length (**I**) of the colon of mice in (**F**) after 7 days of DSS. Cs = *C. scindens*. (**J, K**) Total histopathologic score (**J**) and representative H&E stainings of the colon (**K**) of mice in (**F**) after 7 days of DSS ($n = 10$/group). (**L**) Quantification of FITC dextran in plasma of mice in (**F**) after 7 days of DSS. (**M**) 10-week-old male SPF-Van mice were gavaged daily for 5 days with live *C. scindens* ($10^8$ CFU) ($n = 10$) or vehicle (PBS) ($n = 11$) and experimental colitis was induced by a 7-day treatment with DSS (2.5% in drinking water) followed by 3 days of drinking water (recovery period). (**N, O**) Quantification of EdU$^+$ cells per crypt (**N**) and representative images (**O**) in the colon of mice in M at the end of the experiment (SPF-Van vs SPF-Van + *C. scindens* $P < 0.0001$). Scale bar = 100 μm (**K**) and 50 μm (**O**). Graphs represent mean ± SEM. *n* refers to biological replicates. *P* values (exact values) were calculated using one-way ANOVA followed by Tukey's post hoc correction (**C**), two-way ANOVA followed by Bonferroni's post hoc correction (**G**) or 2-tailed Student's t-test (**D, I, J, L, N**). Source data are available online for this figure.

intestinal barrier function, as indicated by reduced intestinal bacterial translocation to the spleen (Fig. EV2K) and decreased diffusion of orally administered FITC dextran into the blood (Fig. 2L). Furthermore, adding a recovery period at the end of the DSS treatment (Fig. 2M) triggered a regenerative response in the colonic crypts that was more pronounced in the SPF-Van mice with *C. scindens*, as evidenced by EdU$^+$ and Ki67$^+$ staining and quantification (Figs. 2N,O and EV2L,M). Of note, this proliferation phenotype was absent in unchallenged SPF-Van mice (Fig. EV2N,O), suggesting that *C. scindens* triggers only a damage-induced and controlled cell proliferation.

## TGR5-dependence and therapeutic relevance of the *C. scindens*–7α-dehydroxylated BA axis in intestinal epithelial injury

We next investigated whether activation of the BA membrane receptor TGR5 underlies the improved mucosal healing phenotype in *C. scindens*-colonized SPF-Van mice. For this purpose, we subjected 10-week-old male *Tgr5* wild-type (*Tgr5*$^{+/+}$) and *Tgr5* knock-out (*Tgr5*$^{-/-}$) SPF-Van mice to the DSS colitis recovery protocol (Fig. 3A). In line with our hypothesis, *C. scindens* colonization protected *Tgr5*$^{+/+}$, but not *Tgr5*$^{-/-}$, SPF-Van mice from chemically-induced epithelial injury, as indicated by reduced body weight loss (Fig. 3B), increased colon length (Fig. 3C,D), and intestinal cell proliferation (Fig. 3E,F).

To verify in vivo TGR5 activation, we performed transcriptomics analysis (BRB-seq (Alpern et al, 2019)) on the colons of these mice. Gene set enrichment analysis (GSEA) revealed a significant increase in intestinal stem cell (ISC) proliferation-related pathways in *Tgr5*$^{+/+}$ SPF-Van mice colonized with *C. scindens* compared to *Tgr5*$^{+/+}$ SPF-Van control mice, while those pathways were mainly downregulated in *Tgr5*$^{-/-}$ SPF-Van control mice (Fig. 3G,H).

To evaluate whether our observations hold therapeutic potential, we administered *C. scindens* immediately after instead of before the DSS treatment (Fig. EV3A). In this experiment, preconditioning with vancomycin was omitted as DSS is known to induce intestinal dysbiosis (Munyaka et al, 2016), possibly sufficient to free intestinal niches and reduce competition to allow *C. scindens* engraftment.

Importantly, post-DSS administration of *C. scindens* was sufficient to accelerate the recovery, as evidenced by the faster body weight regain (Fig. EV3B) and increased colon length (Fig. EV3C). Notably, an independent experiment comparing isogenic *Tgr5*$^{+/+}$ with *Tgr5*$^{-/-}$ mice revealed that this therapeutic effect required the presence of TGR5 (Fig. 3I–K). In line with our previous study (Sorrentino et al, 2020), *Tgr5*$^{+/+}$ mice colonized with *C. scindens* exhibited increased numbers of Chromogranin A-positive (ChgA$^+$) enteroendocrine cells (EECs), suggesting enhanced differentiation of these cells (Fig. 3L,M). Collectively, these findings imply that introduction of *C. scindens* in the distal gut fosters intestinal epithelial cell (IEC) regeneration and reestablishes homeostasis through TGR5 activation.

## Alteration of 7α-dehydroxylated BA generation is associated with a defect in intestinal cell differentiation in UC patients

To translate our results to humans, we reanalyzed public multi-omics datasets from UC patients and non-IBD individuals (Lloyd-Price et al, 2019). GSEA revealed not only an expected enrichment in proinflammatory pathways in the rectum (Fig. 4A) and colon (Fig. EV4A) of UC patients, but also a reduction in the specification of different intestinal cell types, indicating an impairment in cell differentiation in UC (Figs. 4A,B and EV4A). To assess whether these effects could be mediated by the 7α-dehydroxylated BA-TGR5 signaling, we analyzed the *Tgr5* expression in the human intestine. Examination of RNA-sequencing (RNA-seq) data from ileum, colon, and rectum showed no significant changes in *Tgr5* expression in UC patients compared to non-IBD individuals (Fig. 4C). In addition, correlation between fecal BA composition of non-IBD individuals and UC patients and gene signatures within enterocytes, goblet cells, and EECs revealed a strong positive correlation with unconjugated 7α-dehydroxylated BAs (e.g., DCA, LCA) and a negative correlation with unconjugated (e.g., CA, CDCA) and conjugated (e.g., TCA) primary BAs (Fig. 4D (rectum) and EV4B (colon)). In addition, single-cell deconvolution highlighted a significant reduction in goblet cells and EECs, associated with an increase in tuft cells and fibroblasts in UC patient rectum biopsies (Fig. 4E). Altogether, these observations indicate a

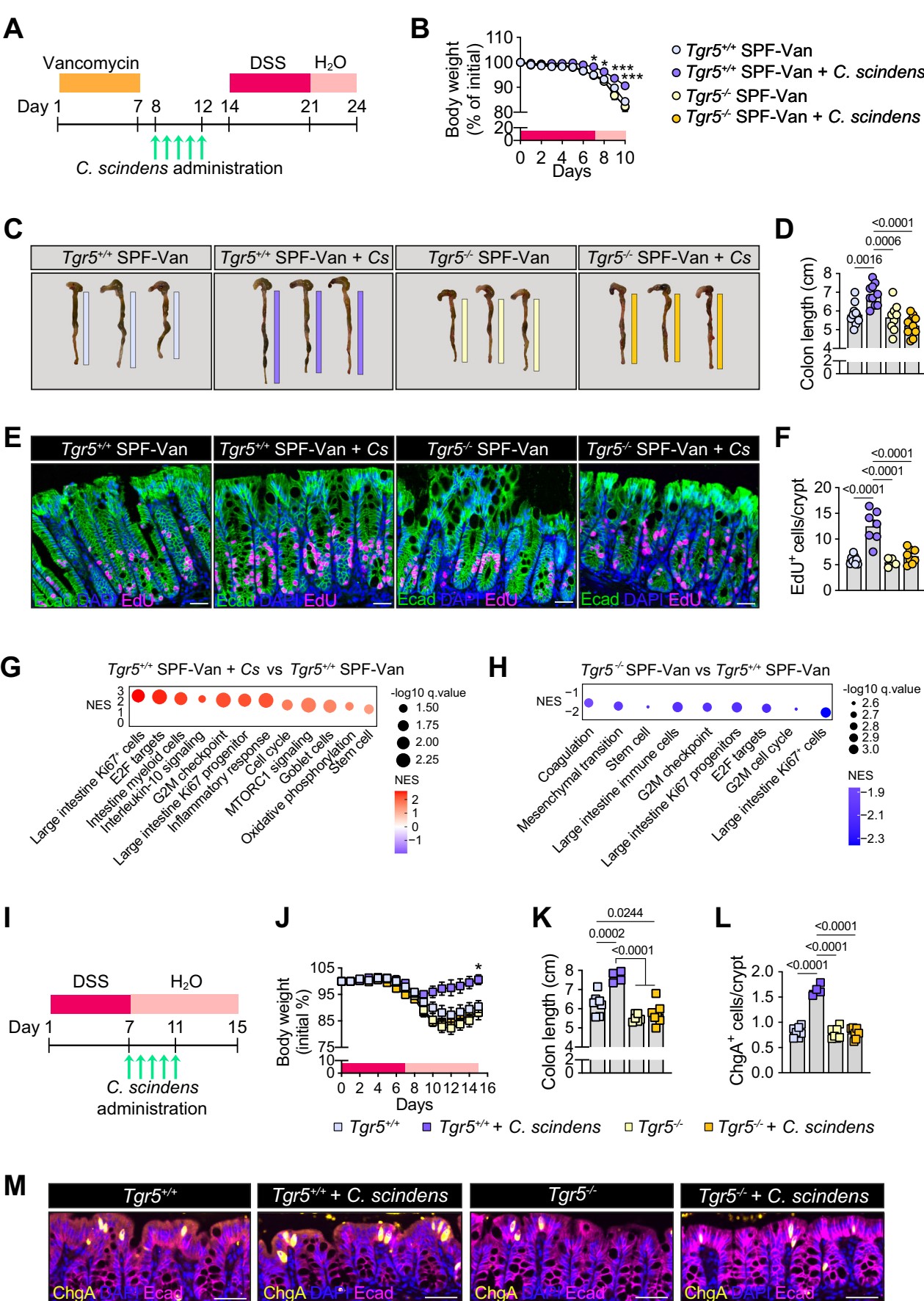

◄ **Figure 3. TGR5-dependent restoration of intestinal homeostasis after experimental colitis-induced mucosal injury.**

(A) 10-week-old male C57BL/6J SPF-Van wild-type ($Tgr5^{+/+}$) and TGR5 knock-out ($Tgr5^{-/-}$) mice were gavaged daily for 5 days with live *C. scindens* ($10^8$ CFU – $Tgr5^{+/+}$ SPF-Van + *C. scindens* and $Tgr5^{-/-}$ SPF-Van + *C. scindens*) ($n = 10$/group) or vehicle (PBS - $Tgr5^{+/+}$ SPF-Van ($n = 11$) and $Tgr5^{-/-}$ SPF-Van ($n = 9$)). Experimental colitis was induced by a 7-day treatment with DSS (2.5% in drinking water) followed by 3 days of drinking water (recovery period). (B) Percentage of body weight loss of mice in (A) ($Tgr5^{+/+}$ SPF-Van vs $Tgr5^{+/+}$ SPF-Van + *C. scindens* day 7: *$P = 0.022$, day 8: *$P = 0.028$, day 9: ***$P = 0.0004$, day 10: ***$P < 0.0001$; $Tgr5^{+/+}$ SPF-Van + *C. scindens* vs $Tgr5^{-/-}$ SPF-Van day 7: *$P = 0.015$, day 8: *$P = 0.004$, day 9: ***$P < 0.0001$, day 10: ***$P < 0.0001$; $Tgr5^{+/+}$ SPF-Van + *C. scindens* vs $Tgr5^{-/-}$ SPF-Van + *C. scindens* day 7: *$P = 0.017$, day 8: *$P = 0.003$, day 9: ***$P < 0.0001$, day 10: ***$P < 0.0001$). (C, D) Representative images (C) and length (D) of the colon of mice in (A) at the end of the experiment ($Tgr5^{+/+}$ SPF-Van + *C. scindens* vs $Tgr5^{-/-}$ SPF-Van + *C. scindens* $P < 0.0001$). (E, F) Representative images (E) and quantification of EdU$^+$ cells per crypt (F) in the colon of mice in (A) at the end of the experiment ($Tgr5^{+/+}$ SPF-Van + *C. scindens* vs all other groups $P < 0.0001$). $Tgr5^{+/+}$ SPF-Van ($n = 9$), $Tgr5^{+/+}$ SPF-Van + *C. scindens* ($n = 7$), $Tgr5^{-/-}$ SPF-Van ($n = 4$) and $Tgr5^{-/-}$ SPF-Van + *C. scindens* ($n = 8$). (G, H) GSEA representing a selection of the most modulated biological processes ordered by normalized enrichment score (NES) in the colon from mice in (A). (I) 10-week-old male $Tgr5^{+/+}$ and $Tgr5^{-/-}$ mice were treated with DSS (2% in drinking water) for 7 days followed by a recovery phase. From the first day of the recovery period, mice were gavaged daily for 5 days with live *C. scindens* ($n = 7$/group except for $n = 4$ for $10^8$ CFU – $Tgr5^{+/+}$ + *C. scindens*). Body weight was monitored daily until one of the four groups reached the initial body weight. (J) Percentage of body weight loss of mice in (I) ($Tgr5^{+/+}$ SPF-Van vs $Tgr5^{+/+}$ SPF-Van + *C. scindens* day 16: *$P = 0.019$; $Tgr5^{+/+}$ SPF-Van + *C. scindens* vs $Tgr5^{-/-}$ SPF-Van day 16: *$P = 0.008$; $Tgr5^{+/+}$ SPF-Van + *C. scindens* vs $Tgr5^{-/-}$ SPF-Van + *C. scindens* day 16: *$P = 0.004$). (K) Colon length of mice in (I) at the end of the experiment ($10^8$ CFU ($Tgr5^{+/+}$ + *C. scindens* ($n = 4$) and $Tgr5^{-/-}$ + *C. scindens* ($n = 7$)) or vehicle (PBS - $Tgr5^{+/+}$ ($n = 7$) and $Tgr5^{-/-}$ ($n = 6$)) ($Tgr5^{+/+}$ SPF-Van + *C. scindens* vs $Tgr5^{-/-}$ SPF-Van + *C. scindens* $P < 0.0001$). (L, M) Quantification of Chromogranin A-positive (ChgA$^+$) cells per crypt (L) and representative images (M) in the colon of mice in (I) at the end of the experiment ($Tgr5^{+/+}$ SPF-Van + *C. scindens* vs all other groups $P < 0.0001$). Scale bar = 50 μm (E, M). Graphs represent mean ± SEM. *n* refers to biological replicates. *P* values (exact values) were calculated using two-way ANOVA followed by Tukey's post hoc correction (B, J) or one-way ANOVA followed by Bonferroni's post hoc correction (D, F, K, L). Source data are available online for this figure.

---

significant association between IEC specification and 7α-dehydroxylated BA levels in UC patients.

## Discussion

Mucosal regeneration and repair following intestinal damage are essential adaptive processes to preserve epithelial barrier integrity and restore intestinal homeostasis. In IBD, however, these processes can be compromised, which, in turn, can further exacerbate inflammation and uncontrolled immune responses (Odenwald and Turner, 2017). In the present study, we explored novel approaches to mitigate intestinal damage in UC by harnessing the metabolic interplay between the gut microbiome and BA physiology. The ratio of secondary-to-primary BAs is altered in human UC, with a significant decrease in 7α-dehydroxylated BAs (Duboc et al, 2013; Lloyd-Price et al, 2019; Sinha et al, 2020). In this study, we demonstrated that colonizing various mouse models with the human-derived 7α-dehydroxylating bacterium *C. scindens* improves the outcomes of experimental colitis. Furthermore, this protective effect was not observed in $Tgr5^{-/-}$ mice, indicating that TGR5 is required for the *C. scindens*-mediated reduction of intestinal damage. Taken together, these results suggest a direct causal relationship between the production of 7α-dehydroxylated BAs by *C. scindens* and the intestinal protective effects elicited by this bacterium. Of note, while *C. scindens* administration did not significantly alter the overall family composition of the gut microbiome ecosystem, we cannot exclude the possibility that this colonization leads to the production of additional, yet unidentified, beneficial metabolites. However, since the shielding effects against DSS phenotype largely depends on the BA receptor, TGR5, our data strongly support the notion that the 7α-dehydroxylated BA-TGR5 axis is a prime mediator of the protective phenotype.

In previous studies, we underscored the importance of TGR5 in driving intestinal regeneration by stimulating ISC proliferation and differentiation (Lund et al, 2020; Sorrentino et al, 2020). While this earlier work uncovered the mechanistic basis by which TGR5 controls ISC function, this new study identifies a window of opportunity to locally activate colonic TGR5 and accelerate mucosal healing, by capitalizing on the innate abilities of BA-modifying gut bacteria. To date, systemic pharmaceutical formulations of TGR5 agonists have not been applied clinically due to gallbladder swelling (Briere et al, 2015; Li et al, 2011) and cardiovascular alterations (Piotrowski et al, 2013; Phillips et al, 2014). As an alternative, modulating the microbial community to restore the in situ production of TGR5 agonists in their appropriate biological compartment and at physiological concentrations may provide a therapeutic strategy that could circumvent these side effects. The protective effect of *C. scindens* in experimental colitis was evident across distinct mouse models, from simplified Oligo-MM$^{12}$ mice to conventional SPF mice with a complex microbiome. For the latter, *C. scindens* gut colonization was successful both in preventive and therapeutic approaches, yet reliant on gut microbiome modulation obtained either by vancomycin preconditioning or DSS administration. This modulation is most likely required to decrease the competition for ecological niches and resources. Our findings in mice align with the results from a human phase 1b safety trial, showing significant improvements in the colonization and effectiveness of a Firmicute-based biotherapeutic product when combined with vancomycin preconditioning to induce remission in patients with mild to moderate UC (Henn et al, 2021).

Finally, our results obtained from publicly available omics datasets of UC and healthy individuals (Lloyd-Price et al, 2019) support the notion that restoring the balance between primary BAs and secondary 7α-dehydroxylated BAs could stimulate regeneration in UC patients. In accordance with the literature (Gersemann et al, 2011), we demonstrated that the differentiation of ISCs into multiple intestinal cell types, including enterocytes, goblet cells, and EECs, is significantly impaired in UC patients. Most notably, the genes in these differentiation pathways showed a robust positive correlation with the levels of 7α-dehydroxylated BAs and a negative correlation with primary and conjugated BAs across all individuals, while the expression of *Tgr5* in the different intestinal segments remains unaffected by UC insurgence.

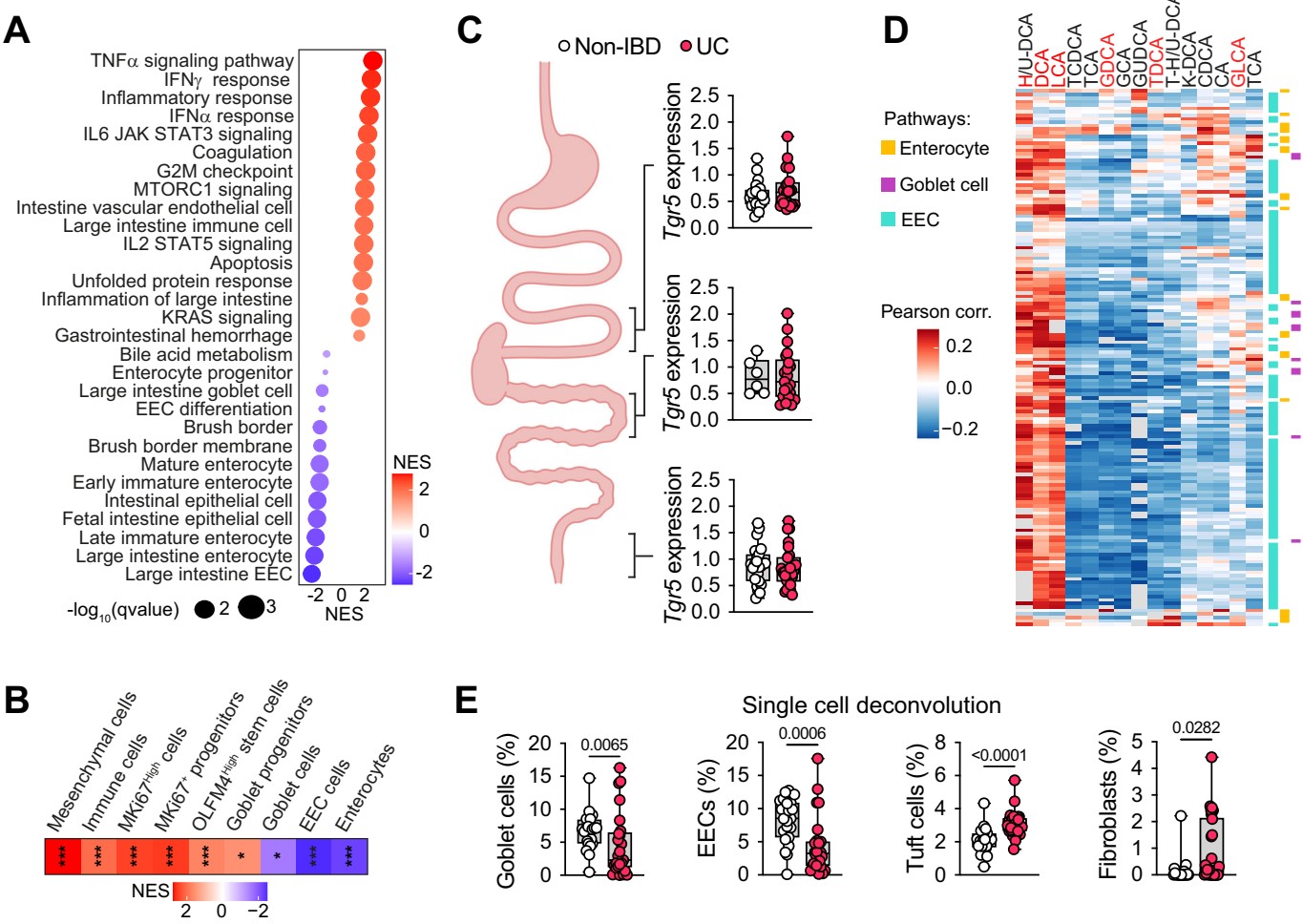

**Figure 4. 7α-dehydroxylated BAs are associated with intestinal cell differentiation in UC patients.**

(A) GSEA representing a selection of the most modulated biological processes in the rectum of ulcerative colitis (UC) patients compared to controls (non-IBD) ($n = 23$ for non-IBD and $n = 26$ for UC) (Lloyd-Price et al, 2019), ordered by normalized enrichment score (NES). (B) Cell type GSEA results from (A). (C) *Tgr5* mRNA expression (count per million) in human intestinal biopsies from non-IBD individuals and UC patients. Ileum ($n = 21$ for non-IBD and $n = 24$ for UC); colon ($n = 6$ for non-IBD and $n = 21$ for UC); rectum ($n = 23$ for non-IBD and $n = 26$ for UC). (D) Pearson correlation between the abundance of fecal BAs (7α-dehydroxylated BAs in red) and rectum gene expression of individuals in (A) for GSEA core enrichment genes of enterocyte, goblet cell and enteroendocrine cell (EEC) signatures. (E) Single-cell deconvolution estimated percentage of cell types in rectum of individuals in (A) (For tuft cells (%): non-IBD vs UC $P < 0.0001$). Graphs represent mean ± SEM. $n$ refers to biological replicates. $P$ values (exact values) were calculated using 2-tailed Student's t-test (E); FDR-corrected $P$ values: *$q < 0.05$; ***$q < 0.001$ (B). Source data are available online for this figure.

Several reports have shown that the BA-TGR5 signaling axis mitigates intestinal inflammation, another hallmark of colitis pathophysiology (Biagioli et al, 2017; Cipriani et al, 2011; Garibay et al, 2019; Sinha et al, 2020). More recent studies, however, have highlighted the role of newly identified low-abundance microbially-derived derivatives of DCA and LCA in modulating immune responses in IBD by activating various other BA receptors. Specifically, derivatives of LCA and DCA—including isoLCA (Paik et al, 2022), isoalloLCA (Hang et al, 2019), 3oxoLCA (Hang et al, 2019) and isoDCA (Campbell et al, 2020)—have been shown to reduce intestinal inflammation by engaging receptors such as RORγt (Hang et al, 2019; Paik et al, 2022), VDR (Song et al, 2020) and FXR (Campbell et al, 2020). These immune-centered studies, along with our findings focused on mucosal

regeneration, underscore the importance of identifying the whole spectrum of the BA metabolizing bacterial species and their enzymatic machinery involved in the fine-tuning of the chief gut host adaptive responses to injury. More studies are needed to dissect how these bacterial species converge and contribute to control local immunity and regeneration in the gut epithelium.

In conclusion, our study reports that *C. scindens* has the ability to restore the equilibrium between 7α-dehydroxylated BAs and their precursors, CA and CDCA, thereby facilitating mucosal healing in mouse models of DSS-induced experimental colitis. Further investigations are necessary to validate these promising findings in other experimental colitis models and explore their potential therapeutic applications for UC patients.

# Methods

### Reagents and tools table

| Reagent/Resource | Reference or Source | Identifier or Catalog Number |
|---|---|---|
| **Experimental models** | | |
| *Clostridium scindens* | Morris et al, 1985 | ATCC 35704 |
| Oligo-MM¹² mice | Brugiroux et al, 2016 | |
| C57BL/6J mice | Charles River, France | |
| *Tgr5⁺/⁺* and *Tgr5⁻/⁻* mice | Thomas et al, 2009 | |
| **Recombinant DNA** | | |
| **Antibodies** | | |
| anti-Ki67 | ThermoFisher | Cat#MA5-14520 |
| anti-Chromogranin-A | Santa Cruz | Cat#sc-13090 |
| anti-E-cadherin | R&D | Cat#AF748 |
| Alexa Fluor conjugated anti-goat A488 | Invitrogen | Cat#A-11029 |
| Alexa Fluor conjugated anti-rabbit A647 | Invitrogen | Cat#A-31573 |
| **Oligonucleotides and other sequence-based reagents** | | |
| *BaiCD_C.scindens* Forward primer | | CTTAAGAACCGTATCGTCCTG |
| *BaiCD_C.scindens* Reverse primer | | CCGGACATAAGGCTACACATT |
| 16S_Forward primer | | ACTCCTACGGGAGGCAGCAG |
| 16S_Reverse primer | | ATTACCGCGGCTGCTGCTGG |
| *Shp* Forward primer | | AGGGCTCCAAGACTTCACACA |
| *Shp* Reverse primer | | CGATCCTCTTCAACCCAGATG |
| *Fgf15* Forward primer | | ACGGGCTGATTCGCTACTC |
| *Fgf15* Reverse primer | | TGTAGCCTAAACAGTCCATTTCCT |
| *Ibap* Forward primer | | TTGAGAGTGAGAAGAATTACGATGAGT |
| *Ibap* Reverse primer | | TTTCAATCACGTCTCCTGGAA |
| *Cyp7a1* Forward primer | | GTCCGGATATTCAAGGATGCA |
| *Cyp7a1* Reverse primer | | AGCAACTAAACAACCTGCCAGTACTA |
| *Bsep* Forward primer | | ACGGTGATGATAGATGGTCACGAC |
| *Bsep* Reverse primer | | TCCACGGAGATCTCTTTGGTGTTG |
| **Chemicals, Enzymes and other reagents** | | |
| Erythromycin | Merck | Cat#E5389 |
| Nalidixic acid | Merck | Cat#N8878 |
| QIAamp Fast DNA Stool Mini Kit | Qiagen | Cat#51604 |
| SYBR Green | Roche | Cat#4887352001 |
| PrimeScript RT Reagent Kit | Takara | Cat#RR047B |

| Reagent/Resource | Reference or Source | Identifier or Catalog Number |
|---|---|---|
| Vancomycin | Teva Pharma | |
| Dextran sulfate sodium salt | Sigma-Aldrich | Cat#42867 |
| FITC-dextran | Sigma-Aldrich | Cat#FD405 |
| 5-ethynyl-2'-deoxyuridine | ThermoFisher Scientific | Cat#A10044 |
| Click-iT™ EdU Alexa Fluor™ 647 | ThermoFisher Scientific | Cat#C10340 |
| Epredia™ Formal-Fixx™ 10% neutral buffered formalin | Fisher Scientific | Cat#9990244 |
| Harris Hematoxylin | Biosystems | Cat#3873.2500 |
| Eosin Y Solution | Sigma | Cat#E4382 |
| Bovine serum albumin (BSA) | Merck | Cat#A7906 |
| Triton X-100 | Merck | Cat#X100 |
| DAPI | ThermoFisher Scientific | Cat#62248 |
| ProLong Gold Antifade Mountant | ThermoFisher Scientific | Cat#P10144 |
| MERCURIUSTM BRB-seq library preparation kit | Alithea Genomics | Cat#10813 |
| Direct-zol-96 RNA | Zymo Research | Cat#R2054 |
| **Software** | | |
| QuPath software | Bankhead, P. et al, 2017 | Version 0.5.1 |
| Rstudio | https://www.r-project.org | R version 4.1.0 |
| *ggplot2* | Wickham H (2016) | |
| *Limma-Voom* | Ritchie et al, 2015 | package version 3.42.2 |
| edgeR calcNormFactors | Robinson et al, 2010 | |
| clusterProfiler | Yu et al, 2012 | |
| GraphPad Prism 10 | Graphpad software | Version 10.0.0 |
| DADA2 pipeline | Callahan et al, 2016 | |
| Phyloseq R package | McMurdie and Holmes, 2013 | |
| **Other** | | |
| Chow diet Oligo-MM¹² experiments | Kliba-Nafag | 3307 |
| Chow diet | DS-SAFE | 150 |
| Anaerobic chamber | Coy Laboratory Products | 95% N₂, 5% H₂ |
| Anaerobic chamber | Don Whitley Scientific A45 HEPA | 80% N₂, 10% CO₂, 10% H₂ |
| LightCycler 480 Real-Time PCR System | Roche | |

| Reagent/Resource | Reference or Source | Identifier or Catalog Number |
|---|---|---|
| SpectraMax ID3 | Molecular Devices | |
| Microm HM325 | ThermoFisher | |
| Slide scanner | Olympus | VS120-L100 |
| 2100 Bioanalyzer | Agilent | |
| Precellys 24 Tissue Homogenizer | Bertin Instruments | |

## Mouse experiments

Mice were housed with ad libitum access to water and food and kept under a 12 h dark/12 h light cycle with a temperature of $22\,°C \pm 1\,°C$ and a humidity of $60\% \pm 20\%$. Mice were euthanized in the evening following 2 h of physiological feeding after a 12 h light-phase fasting.

## Ethics approval

All mouse experiments were authorized by the Veterinary Office of the Canton of Vaud and the Canton of Bern, Switzerland, under the license authorizations no. 3263.1., no. 3917 and no. BE66/2019, respectively. All studies complied with ethical standards.

## *Clostridium scindens* cultivation

*C. scindens* was grown anaerobically in Brain Heart Infusion Supplement – Salts (BHI-S), consisting of 37 g brain heart infusion, 5 g yeast extract, 40 mL salt solution (0.2 g $CaCl_2$, 0.2 g $MgSO_4$, 1 g $K_2HPO_4$, 1 g $KH_2PO_4$, 10 g $NaHCO_3$, and 2 g NaCl in 1 L $ddH_2O$), 1g L-cysteine and 2 g fructose per L $ddH_2O$. For selective *C. scindens* colony-forming unit (CFU) quantification by plating, BHIA-S agar was supplemented with erythromycin (20 μg/mL) and nalidixic acid (50 μg/ml). Bacterial cultures were incubated at $37\,°C$ in an anaerobic chamber.

## Evaluation of *C. scindens* colonization efficiency

DNA was extracted from fresh feces using QIAamp Fast DNA Stool Mini Kit. *C. scindens* abundance was analyzed using the LightCycler 480 Real-Time PCR System and SYBR Green chemistry. Specific primers (listed in the Reagents and tools table) for the *C. scindens BaiCD* gene were used to evaluate *C. scindens* abundance. *C. scindens* abundance was normalized on 16S rRNA copy number.

## Colonization of gnotobiotic mice with *C. scindens*

Gnotobiotic Oligo-MM[12]-associated mice have been established at the Clean Mouse Facility of the University of Bern by inoculation of germ-free C57BL/6J mice with pure cultures of 12 fully genome-sequenced, openly available (DSMZ German Collection of Microorganisms and Cell Cultures) murine intestinal bacterial strains known as the Oligo-MM[12] community (Brugiroux et al, 2016) and stably maintained in flexible film isolators under strict axenic conditions (Yilmaz et al, 2021). The Oligo-MM[12] community is BA 7α-dehydroxylation deficient, which can be compensated by additional colonization with *C. scindens* strain ATCC 35704 (Studer et al, 2016). 7-week-old Oligo-MM[12] male mice maintained on autoclavable rodent chow diet were colonized by a single gavage with $10^7$ CFU of live *C. scindens* or vehicle (PBS).

## Colonization of SPF mice with *C. scindens*

Eight- to twelve-week-old C57BL/6J, *Tgr5*[+/+] or *Tgr5*[−/−] (Thomas et al, 2009) male mice fed chow diet (DS-SAFE 150) were given daily oral (gavage) live *C. scindens* bacteria at 3 different concentrations ($10^8$, $10^9$, or $10^{10}$ CFU) or vehicle (PBS) for 15 days or preconditioning with vancomycin (500 mg/L in drinking water) for 7 days (SPF-Van) and gavaged daily for 5 days with live *C. scindens* bacteria at the above CFUs.

## Experimental colitis and regeneration assay

Experimental colitis was induced by administering dextran sulfate sodium salt (DSS – concentration specified in the figure legends) in drinking water. Changes in body weight were monitored on a daily basis and mice were euthanized 7 days after DSS administration. To monitor epithelial regeneration in DSS-treated mice, DSS was replaced with water for 3 days. To monitor the effect of *C. scindens* during the recovery period after 7 days from DSS initiation, mice were euthanized when one of the experimental groups regained its initial body weight.

## FITC dextran intestinal permeability assay

Mice were gavaged with 4 kDa FITC-dextran at 600 mg/kg 4 h before sacrifice. FITC-dextran concentration in plasma was measured by fluorometry at excitation wavelength of 485 nm and emission wavelength of 535 nm using SpectraMax ID3.

## Measurement of translocated bacteria in the spleen

Spleens were collected during euthanasia and crushed in PBS using 70 μm strainers. Homogenates were then seeded in agar plates and incubated at $37\,°C$. The number of CFUs was then counted after 24 h of incubation.

## Intestinal proliferation assay

5-ethynyl-2'-deoxyuridine (EdU) was resuspended in phosphate-buffered saline (PBS), and 200 μL of solution was injected intraperitoneally (50 μg per g of mouse weight) 2 h before euthanasia. Cell proliferation was assessed by EdU assay (Click-iT™ EdU Alexa Fluor™ 647) following manufacturer's instructions.

## Histology

Intestinal tissues were Swiss rolled, fixed with Epredia™ Formal-Fixx™ 10% neutral buffered formalin overnight at $4\,°C$, and embedded in paraffin. Four sections were prepared by microtome. Hematoxylin and Eosin (H&E) and stainings were performed according to the manufacturer's protocols. Images were acquired using Olympus slide scanner.

*Histopathologic scoring*

A European board-certified veterinary pathologist performed the histopathologic evaluation in a blinded fashion. The following parameters were identified: (1) severity of inflammation; (2) ulceration; (3) crypt damage. Histologic criteria for each parameter have been adopted and adapted (Erben et al, 2014). The sum of the 3 parameter values was used to generate a "total histopathologic score".

## Immunohistochemistry and immunofluorescence

Antigen retrieval was performed by incubating the colon sections in 10 mM citrate buffer (pH 6.0) for 20 min at 95 °C. After cooling to room temperature, the sections were washed and blocked with blocking buffer (1% BSA and 0.5% Triton X-100 in PBS) for 1 h at room temperature. For immunofluorescence, the primary antibodies anti-Ki67, anti-Chromogranin-A and anti-E-cadherin were diluted 1:100 in blocking buffer and incubated overnight at 4 °C. Sections were washed and incubated for 1 h with Alexa Fluor conjugated secondary antibodies (1:1000 in blocking solution). Following extensive washing, sections were counterstained with DAPI and mounted in ProLong Gold Antifade Mountant. Stained sections were imaged by a virtual slide microscope. Image analysis was performed using QuPath software.

## 16S rRNA sequencing and analysis

DNA was extracted from fresh mouse feces using a stool DNA kit, and 16S rRNA sequencing was performed (BGI, China). Briefly, the 16S rRNA V3-V4 region was amplified using 16S rRNA fusion primers. All PCR products were purified by Agencourt AMPure XP beads, dissolved in elution buffer, and labeled for library construction. Library size and concentration were detected by Agilent 2100 Bioanalyzer. Qualified libraries were sequenced on HiSeq platform according to their insert size. Amplicon sequence variants and taxonomy assignment were generated using DADA2 pipeline (Callahan et al, 2016) and microbiome data analysis was performed using Phyloseq R package (McMurdie and Holmes, 2013).

## BA quantification

To extract the BAs listed in Table EV5, frozen feces were lyophilized overnight at −60 °C. The dried fecal samples were homogenized to powder. 50 mg of each sample was weighed and 6 ceramic beads (2.5 mm) were added to each tube. For plasma, BAs were extracted from 30 µl. 1500 µL of MeOH/H$_2$O (2/1) + 0.1% formic acid was used as the extraction solvent. Samples were homogenized in a Precellys 24 Tissue Homogenizer at 6500 rpm 2 × 20" beat and 20" rest. The homogenized samples were centrifuged at 21,000 rcf, for 15 min, at 4 °C. Samples (100 µL of fecal sample or 40 µL of plasma) and 100 µL of calibration standard mix were transferred to individual wells of 2 mL 96-well plate. 50 µL of an internal standard spiking (ISTD) solution (CA-d4, CDCA-d4, TCA-d4, TUDCA-d4, DCA-d4 and LCA-d4, each at 2 µM in MeOH) was amended to each well. Immediately after the addition of ISTD, 600 µL of 0.2% formic acid in H$_2$O was added to each sample or calibration standard level. The 96-well plate was shaken with an orbital shaker at 300 rpm and centrifuged at 3500 rpm, 5 min, 4 °C. The contents of the 96-well plate were extracted by solid phase extraction with an Oasis HLB 96-well

µElution plate. The extracted samples were dried in a Biotage® SPE Dry 96 at 20 °C and reconstituted with 100 µL of MeOH/H$_2$O (50/50). The plate was shaken with an orbital shaker at 300 rpm, 5 min and centrifuged at 3500 rpm, 5 min, 4 °C. Liquid chromatography-mass spectrometry (LC-MC) was performed as previously reported (Vico-Oton et al, 2022).

## Hydrophobicity index calculation

Hydrophobicity index was calculated using the BA hydrophobicity scores as described previously (Heuman, 1989) using the formula: HI = $\sum C_i / \sum (C_i \times H_i)$. $C_i$: concentration of the BA species; $H_i$: hydrophobicity score of the BA species; $\sum C_i$: total concentration of all the BAs in the plasma.

## Fecal 7α-dehydroxylating activity measurement

Feces were freshly collected and resuspended in PBS before being incubated in BHI-S containing 100 µM of CA for 24 h at 37 °C in an anaerobic chamber. BA were extracted from the culture and the 7α-dehydroxylating activity, reflected by the conversion of CA to DCA, was normalized to the protein content of the fecal bacterial culture. Each replicate consists of fecal pellets pooled from four different mice.

## RNA extraction and quantitative real-time qPCR

RNA was extracted from biobanked liver and terminal ileum using Direct-zol-96 RNA and transcribed to complementary DNA using PrimeScript RT Reagent Kit following manufacturer's instruction. Expression of the indicated genes (primers listed in the Reagents and tools table) was analyzed using the LightCycler 480 Real-Time PCR System and SYBR Green chemistry. All the quantitative polymerase chain reaction (PCR) results were presented relative to the mean of cyclophilin housekeeping gene (DDCt method). The average of three technical repeats was used for each biological data point.

## Bulk RNA barcoding and sequencing (BRB-seq)

Samples were sent to Alithea Genomics SA (Lausanne, Switzerland) for library preparation and sequencing using highly multiplexed 3′-end bulk RNA barcoding and sequencing (MERCURIUSTM BRB-seq service). The generation of bulk RNA Barcoding and sequencing (BRB-seq) libraries was performed using the MERCURIUSTM BRB-seq library preparation kit for Illumina and following the manufacturer's manual. All libraries were sequenced on an Illumina Novaseq 6000. Read trimming, alignment, and quantification steps were performed by Alithea Genomics using STARsolo. R2 reads were trimmed ("--clipAdapterType CellRanger4") and aligned to the GRCm38 genome. To generate both raw and UMI-deduplicated counts, the following parameter was used: "--soloUMIdedup NoDedup 1MM_Directional". UMI-deduplicated count matrices were used for downstream analyses.

## Estimation of cell type proportions

To estimate the cellular composition of the human large intestine samples, we performed single-cell deconvolution using MuSiC version 0.2.0 (Wang et al, 2019) on raw bulk RNA-seq counts using the default parameters and a maximum number of iteration equal to 1500. We retrieved large intestine FACS single-cell RNA-seq

**The paper explained**

**Problem**

Ulcerative colitis (UC) remains uncontrolled in approximately 25% of cases despite advances in therapy, often necessitating immunosuppressive drugs with significant risks, such as infections and malignancies. Promoting mucosal healing after relapse resolution is a promising alternative approach, offering enhanced tissue repair and sustained remission while minimizing the risks associated with immunosuppression.

**Results**

We developed a live biotherapeutic product using *Clostridium scindens*, a bile acid 7α-dehydroxylating bacterium that stimulates secondary bile acid production. In mouse models of dextran sulfate sodium-induced acute experimental colitis, *Clostridium scindens* administration reduced disease severity, maintained colon integrity, and improved intestinal barrier function, driven by enhanced intestinal regeneration and mucosal healing.

**Impact**

Our study highlights *Clostridium scindens* as a promising biotherapeutic to restore bile acid homeostasis and promote mucosal healing following colon injury. Further studies are required to validate these findings in other experimental colitis models and translate them into therapeutic applications for UC patients.

processed counts and samples annotations from the Tabula Sapiens consortium (Tabula Sapiens Consortium et al, 2022).

### BRB-seq and RNA-seq downstream analysis

Differential expression was performed using Limma-Voom (Ritchie et al, 2015) on normalized counts computed with edgeR calcNormFactors (Robinson et al, 2010). The significance threshold was set at 5% after Benjamini-Hochberg multiple testing correction. Gene set enrichment analysis (GSEA) was performed on differentially expressed genes with clusterProfiler (Yu et al, 2012).

### Human UC RNA-seq analysis

Publicly available bulk RNA-seq processed counts from the indicated part of the intestine of UC patients (21 colonic biopsies and 26 rectal biopsies) and controls (Non-IBD) (5 colonic biopsies and 21 rectal biopsies) were downloaded from the Inflammatory Bowel Disease Multi'omics database (www.ibdmdb.org) associated with the BioProject number PRJNA398089 (Lloyd-Price et al, 2019). *Tgr5* mRNA expression was represented as count per million (CPM) extracted from RNA-seq data.

### Study design and statistical analysis

This study was carried out in compliance with the ARRIVE guidelines regarding the use of animals in research. Mice were assigned to the different groups based on their genotypes and body weight to ensure that the mean body weights across the groups remained constant. The sample size was determined by the known variability for each assay, and a power analysis was conducted to calculate the appropriate sample size for the mouse experiments. Mice displaying any signs of distress, as predefined in the animal licenses, were euthanized and excluded from the study.

All experimental procedures, including sample collection, data acquisition, and analysis, were performed in a blinded manner.

Statistically significant differences between the means of two groups were assessed by unpaired $t$ test, one- or two-way analysis of variance as specified in the legends. All statistical analyses were calculated using GraphPad Prism 9 software. A $P$ value of 0.05 or less was considered statistically significant.

## Data availability

The dataset produced in this study are available in the following databases: 16S (Sequence Read Archive accession number: PRJNA1209827). BRB-seq (Gene Expression Omnibus accession number: GSE287080).

The source data of this paper are collected in the following database record: biostudies:S-SCDT-10_1038-S44321-025-00202-w.

## Peer review information

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

## Acknowledgements

We thank Giacomo von Alvensleben, the EPFL Center of PhenoGenomics and the EPFL Histology Core Facility for technical assistance. Gnotobiotic animal experimentation was supported by the Clean Mouse Facility of the University of Bern and the Genaxen Foundation. SH and YD thank Mohana Mukherjee, Hai Li, Luca Beldi, Karin Stettler, and Jorum Kirundi for experimental support. This work was funded by the Ecole Polytechnique Fédérale de Lausanne (EPFL) and the Swiss National Science Foundation (SNSF N° 310030_189178 to KS and Sinergia CRSII5_180317/1 to RBL, SH, and KS).

## Author contributions

**Antoine Jalil**: Conceptualization; Data curation; Software; Formal analysis; Validation; Investigation; Visualization; Methodology; Writing—original draft; Project administration; Writing—review and editing. **Alessia Perino**: Conceptualization; Data curation; Formal analysis; Supervision; Funding acquisition; Investigation; Visualization; Methodology; Writing—original draft; Project administration; Writing—review and editing. **Yuan Dong**: Data curation; Formal analysis; Validation; Investigation. **Jéromine Imbach**: Validation; Investigation. **Colin Volet**: Investigation. **Eduard Vico-Oton**: Methodology. **Hadrien Demagny**: Resources; Supervision; Funding acquisition; Writing—review and editing. **Lucie Plantade**: Resources; Supervision; Funding acquisition; Investigation; Writing—review and editing. **Hector Gallart-Ayala**: Resources; Supervision; Funding acquisition; Investigation; Writing—original draft; Writing—review and editing. **Julijana Ivanisevic**: Funding acquisition; Methodology. **Rizlan Bernier-Latmani**: Resources; Supervision; Funding acquisition; Investigation; Writing—review and editing. **Siegfried Hapfelmeier**: Resources; Supervision; Funding acquisition; Investigation; Writing—review and editing. **Kristina Schoonjans**: Resources; Supervision; Funding acquisition; Methodology; Writing—original draft; Writing—review and editing.

Source data underlying figure panels in this paper may have individual authorship assigned. Where available, figure panel/source data authorship is listed in the following database record: biostudies:S-SCDT-10_1038-S44321-025-00202-w.

## Disclosure and competing interests statement

AJ, AP, and KS are co-inventors of the patent application AG1551EP. The other authors declare no competing interests.

# Expanded View Figures

**Figure EV1.   Stable colonization of Oligo-MM¹² mice with *C. scindens* modulates the BA composition and promotes proliferation of colonic stem cells under basal conditions.**

(A) Schematic illustrating the 7α-dehydroxylation reaction at carbon 7 (highlighted in red) of CDCA to form LCA. (B) Violin plots showing the amount of the indicated BA species in the feces of 8-week-old male Oligo-MM¹² mice gavaged with live *C. scindens* ($10^7$ CFU – Oligo-MM¹² + *C. scindens*) or PBS (control – Oligo-MM¹²) under basal conditions ($n = 7$/group) (For LCA (nmol/g): Oligo-MM¹² vs Oligo-MM¹² + *C. scindens* $P < 0.0001$). (C) Dot plots representing the BA proportion and the z score in stools of mice in (B). The dot color represents the z-score for each BA species, whereas the dot size represents the proportion of each BA species over the total BA amount per mouse. 7α-dehydroxylated BAs are highlighted in red. Ctrl = Oligo-MM¹² and Cs = Oligo-MM¹² + *C. scindens*. (D) Plasma BA composition of mice in (B). BA data are normalized to the total amount of BAs. 7α-dehydroxylated BAs are in red. (E, F) Concentrations (nM) of the indicated BAs and secondary (SBA)-to-primary (PBA) ratio in plasma of mice in (B). 7α-DH-ed: 7α-dehydroxylated BAs. (G) BA hydrophobicity index in plasma of mice in (B). (H, I) Relative mRNA levels of the indicated genes in liver (H) or ileum (I) of mice in (B). (J, K) Representative images (J) and quantification of EdU⁺ cells per crypt (K) in the jejunum, ileum, and colon of mice in (B). Graphs represent mean ± SEM. $n$ refers to biological replicates. $P$ values (exact values) were calculated using 2-tailed Student's t test (B, E, F, G, K).

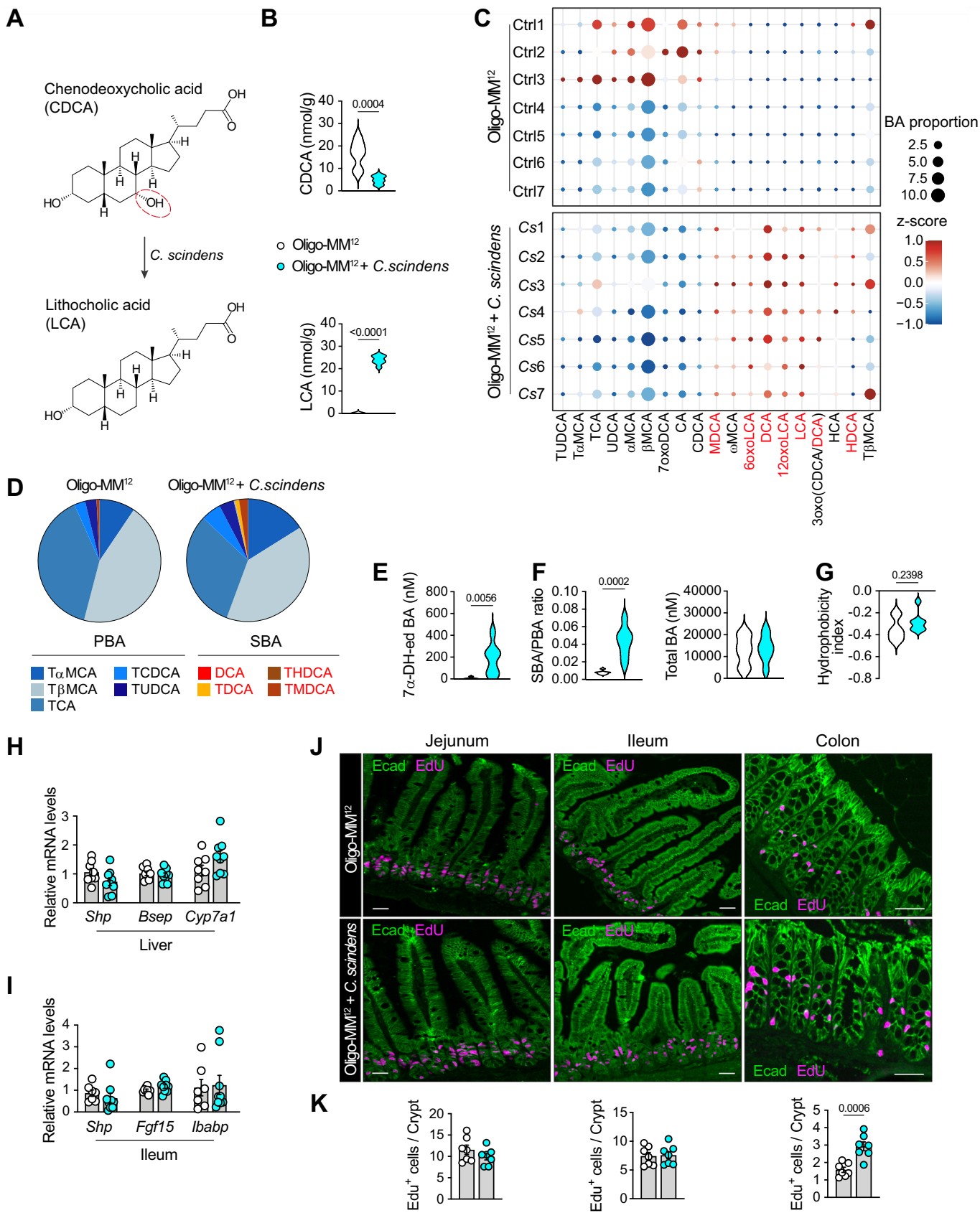

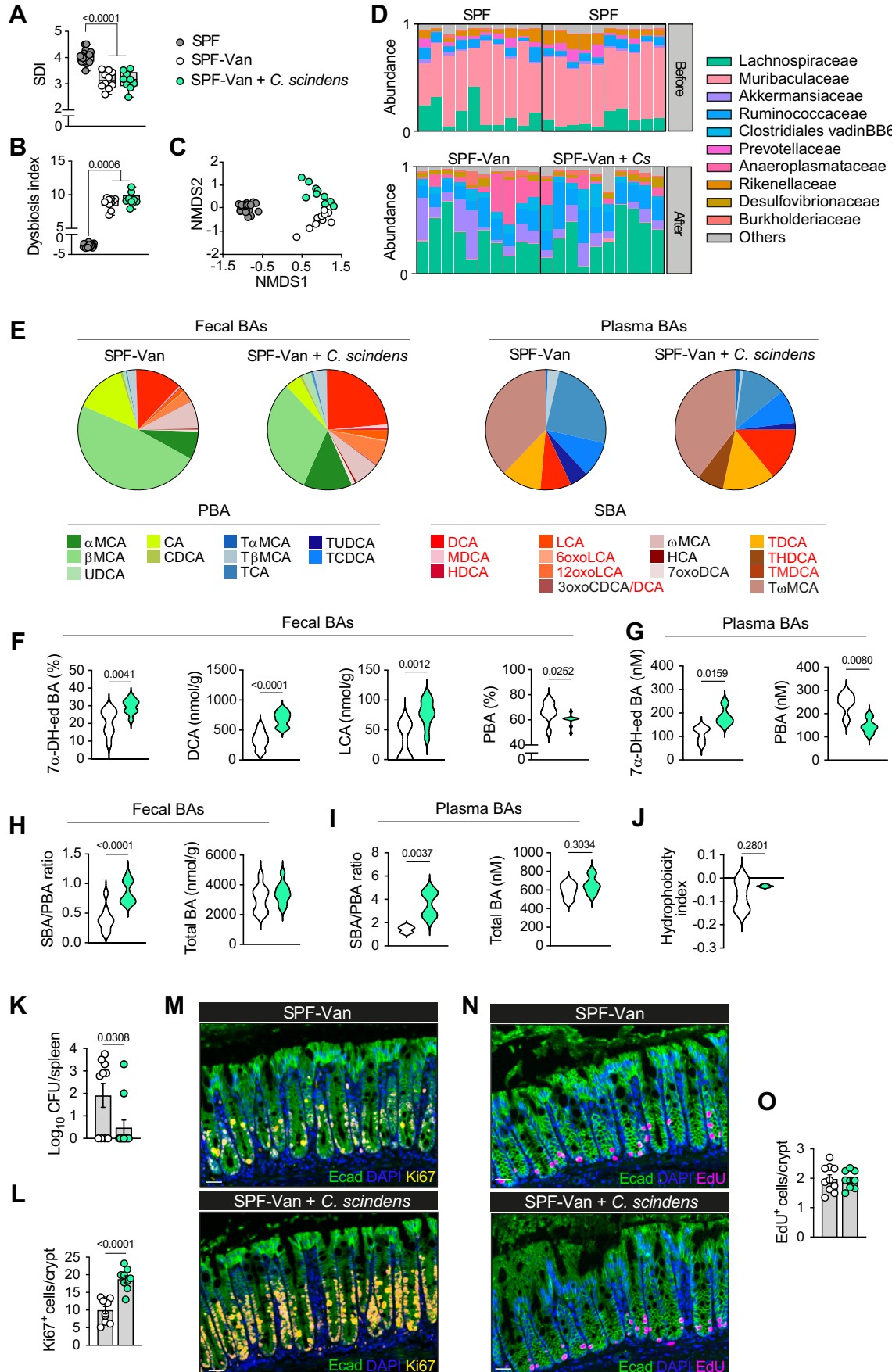

◀ **Figure EV2. Characterization of *C. scindens* colonization in SPF-Van mice.**

(A) Shannon diversity index (SDI) showing the fecal bacterial community diversity of C57BL/6J mice (SPF) ($n = 19$) or preconditioned with vancomycin and gavaged daily for 5 days with live *C. scindens* ($10^8$ CFU – SPF-Van + *C. scindens*) ($n = 9$) or PBS vehicle (SPF-Van) ($n = 10$) (SPF vs all other groups $P < 0.0001$). (B) Gut dysbiosis index of mice in (A). (C) Non-metric multidimensional scaling (NMDS) of amplicon sequence variant (ASV) counts based on the Bray-Curtis dissimilarity of mice in (A). (D) Relative abundance of bacterial families representing more than 10% of total abundance of mice in (A). (E) BA composition in feces ($n = 9$ for SPF-Van and $n = 10$ for SPF-Van + *C. scindens*) and plasma ($n = 5$ for SPF-Van and $n = 4$ for SPF-Van + *C. scindens*), respectively. BA data are normalized to the total amount of BAs. 7α-dehydroxylated BAs are in red. (F) Violin plots showing BA amount (nmol/g of feces) or proportion of the indicated BAs to total BA fecal amount of mice in (E) (For DCA (nmol/g): SPF-Van vs SPF-Van + *C. scindens* $P < 0.0001$). (G) Violin plots showing the BA concentration (nM) of the indicated BAs in the plasma of mice in (E). (H) Secondary (SBA)-to-primary (PBA) BA ratio and total BA amount in feces of mice in (E) (For SBA/PBA ratio: SPF-Van vs SPF-Van + *C. scindens* $P < 0.0001$). (I) Secondary (SBA)-to-primary (PBA) BA ratio and total BA concentration in plasma of mice in (E). (J) BA hydrophobicity index in plasma of mice in (E). (K) Bacterial colony-forming units (CFU) in the spleen of 10-week-old male SPF-Van mice gavaged daily for 5 days with live *C. scindens* ($10^8$ CFU – SPF-Van + *C. scindens*) ($n = 11$) or PBS (control – SPF-Van) ($n = 10$) and subjected to a 7-day treatment with DSS (2.5% in drinking water) (Fig. 2F). (L, M) Quantification (L) and representative images (M) of Ki67+ cells per crypt in the colon of 10-week-old male SPF-Van mice gavaged daily for 5 days with live *C. scindens* ($10^8$ CFU – SPF-Van + *C. scindens*) ($n = 9$) or vehicle (PBS – SPF-Van) ($n = 10$) (SPF-Van vs SPF-Van + *C. scindens* $P < 0.0001$). 2 days after the colonization, experimental colitis was induced by a 7-day treatment with DSS (2.5% in drinking water) followed by 3 days of drinking water (recovery period) (Fig. 2M). (N, O) Representative images (N) and quantification of EdU+ cells per crypt (O) in the colon of unchallenged mice in (A) at the time of sacrifice. Scale bar = 50 μm (M, N). Graphs represent mean ± SEM. *n* refers to biological replicates. *P* values (exact values) were calculated using one-way ANOVA followed by Bonferroni's post hoc correction (A, B) or 2-tailed Student's t test (F, G, H, I, K, L).

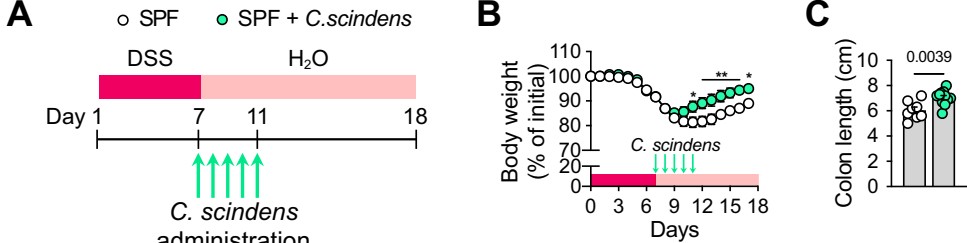

**Figure EV3. Therapeutic effect of *C. scindens* administration in SPF mice.**

(A) Workflow of the therapeutic DSS-induced acute experimental colitis experiment. 10-week-old male C57BL/6J mice were treated with DSS (2.5% in drinking water) for 7 days followed by a recovery phase. From the first day of the recovery period, mice were gavaged daily for 5 days with live *C. scindens* ($10^8$ CFU – SPF + *C. scindens* ($n = 11$)) or vehicle (PBS – SPF ($n = 9$)). Mouse body weight was monitored daily until one of the four experimental groups reached the initial body weight. (B) Percentage of body weight loss of mice in (A) during exposure to DSS and the recovery phase (SPF-Van vs SPF-Van + *C. scindens* day 11: *$P = 0.02$; day 12: **$P = 0.003$; day 13: **$P = 0.002$; day 14: **$P = 0.003$; day 15: **$P = 0.004$; day 16: **$P = 0.004$; day 17: *$P = 0.04$). (C) Colon length of mice in (A) at the end of the experiment. Graphs represent mean ± SEM. $n$ refers to biological replicates. $P$ values (exact values) were calculated using two-way ANOVA followed by Bonferroni's post hoc correction (B) or 2-tailed Student's t-test (C).

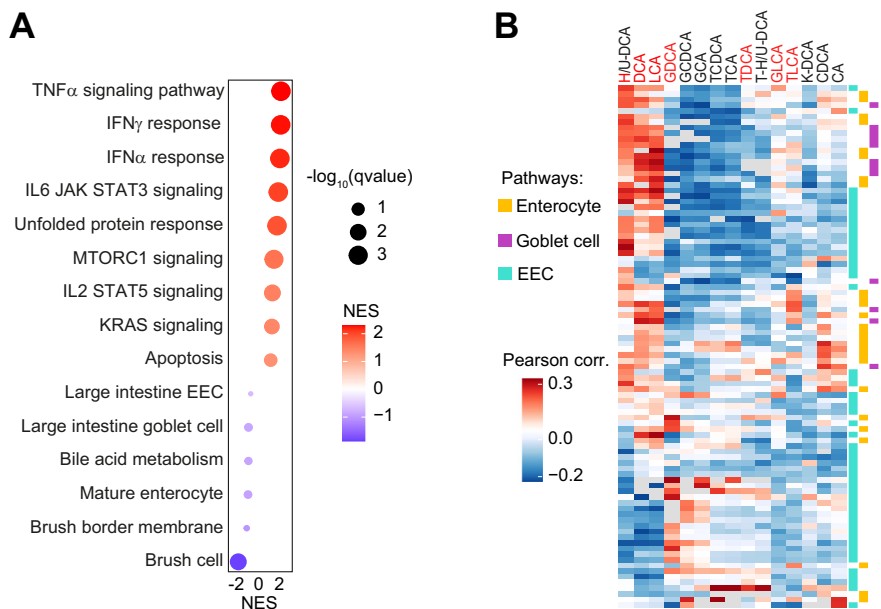

**Figure EV4.  7α-dehydroxylated BAs are associated with intestinal cell differentiation in the colon of UC patients.**

(A) Gene set enrichment analysis (GSEA) representing a selection of the most modulated biological processes in the colon of UC patients compared to non-IBD controls (publicly available dataset from (Lloyd-Price et al, 2019)), ordered by normalized enrichment score (NES). (B) Heatmap representing the correlation (Pearson correlation) between the abundance of different fecal BA species and the genes most significantly involved in biological processes of enterocytes, goblet cells and enteroendocrine cells (EEC) of colon in UC patients and non-IBD controls in (A). 7α-dehydroxylated BAs are highlighted in red.

