## [Peer Review File · EMBO Molecular Medicine]

Bile acid 7 α -dehydroxylating bacteria accelerate injury-induced mucosal healing in the colon

Antoine Jalil, Alessia Perino, Yuan Dong, Jérôme Imbach, Colin Volet, Eduard Vico-Oton, Hadrien Demagny, Lucie Plantade, Hector Gallart-Ayala, Julijana Ivanisevic, Rizlan Bernier-Latmani, Siegfried Hapfelmeier, and Kristina Schoonjans

Corresponding author(s): Kristina Schoonjans (kristina.schoonjans@epfl.ch)

Review Timeline:

Submission Date:	29th Aug 24
Editorial Decision:	15th Oct 24
Revision Received:	24th Jan 25
Editorial Decision:	30th Jan 25
Revision Received:	3rd Feb 25
Accepted:	5th Feb 25

Editor: Jingyi Hou

Transaction Report:

15th Oct 2024

Dear Prof. Schoonjans,

Thank you again for submitting your work to EMBO Molecular Medicine. We have now received feedback from the three referees who agreed to evaluate your manuscript. As you will see in the reports below, the referees find your study of interest but have raised several concerns that will need to be thoroughly addressed in a major revision of the manuscript.

The referees' recommendations are clear, so I won't repeat the points listed below. It's important to carefully address all the issues raised by the referees. Referee #2 mentioned the limitation of relying solely on the DSS model. While including data from additional models is encouraged if such data is already available, it is not required for the manuscript's acceptance. However, potential limitations and future directions in this area need to be discussed.

We would welcome the submission of a revised version within three months for further consideration. As you may already know, our editorial policy allows in principle a single round of major revision, and it is therefore essential to provide responses to the referees' comments that are as complete as possible.

Please also contact us as soon as possible if similar work is published elsewhere. If other work is published, we may not be able to extend the revision period beyond three months.

I look forward to receiving your revised manuscript.

Kind regards,
Jingyi

Jingyi Hou
Editor
EMBO Molecular Medicine

We require:

2) Individual production quality figure files as .eps, .tif, .jpg (one file per figure). For guidance, download the 'Figure Guide PDF': (<https://www.embopress.org/page/journal/17574684/authorguide#figureformat>).

3) A .docx formatted letter INCLUDING the reviewers' reports and your detailed point-by-point responses to their comments. As part of the EMBO Press transparent editorial process, the point-by-point response is part of the Review Process File (RPF), which will be published alongside your paper.

4) A complete author checklist, which you can download from our author guidelines (<https://www.embopress.org/page/journal/17574684/authorguide#submissionofrevisions>). Please insert information in the checklist that is also reflected in the manuscript. The completed author checklist will also be part of the RPF.

5) Please note that all corresponding authors are required to supply an ORCID ID for their name upon submission of a revised

manuscript.

6) It is mandatory to include a 'Data Availability' section after the Materials and Methods. Before submitting your revision, primary datasets produced in this study need to be deposited in an appropriate public database, and the accession numbers and database listed under 'Data Availability'. Please remember to provide a reviewer password if the datasets are not yet public (see <https://www.embopress.org/page/journal/17574684/authorguide#dataavailability>).

12) Author contributions: You will be asked to provide CRediT (Contributor Role Taxonomy) terms in the submission system. These replace a narrative author contribution section in the manuscript.

13) A Conflict of Interest statement should be provided in the main text.

14) Every published paper now includes a 'Synopsis' to further enhance discoverability. Synopses are displayed on the journal

webpage and are freely accessible to all readers. They include a short stand first (maximum of 300 characters, including space) as well as 2-5 one-sentences bullet points that summarizes the paper. Please write the bullet points to summarize the key NEW findings. They should be designed to be complementary to the abstract - i.e. not repeat the same text. We encourage inclusion of key acronyms and quantitative information (maximum of 30 words / bullet point). Please use the passive voice. Please attach these in a separate file or send them by email, we will incorporate them accordingly.

15) Include a Reagents and Tools Table as part of the Methods section, which can be downloaded from our author guidelines (<https://www.embopress.org/page/journal/17574684/authorguide#structuredmethods>)

***** Reviewer's comments *****

Referee #1 (Comments on Novelty/Model System for Author):

State of the art methods and mouse models

Referee #1 (Remarks for Author):

I read this manuscript with extreme interest. The authors present elegant data with state-of-the-art models that connects intestinal bacteria with bile acid metabolism, intestinal mucosa regeneration and inflammation. The data are well presented and the discussion is very well balanced. Nevertheless, few information are needed to complete the story.

1. The authors should clearly present the bile acid profile both in the feces and in the serum or bile and discuss the modulation of the primary/secondary as well as the changes in the hydrophobicity index (which is a key point per se for prevention of intestinal damage)
2. Feces should eventually be studied for the 7- α -dehydroxylated activity (cell reports 2014; 7(1): 12.8). It is important to prove that in the aqueous phase of the feces there is a net 7 α -DH activity on bile acids
3. Gene expression profile is needed for both intestine and liver to prove if modulation of bile acid species as well as hydrophobicity and quantity (pool) drives changes in the classical pathways (SHP, IBABP and FGF15 for the intestine and SHP, BSEP and CYP7A1 for the liver)

Referee #2 (Comments on Novelty/Model System for Author):

DSS only is the primary limitation.

Referee #2 (Remarks for Author):

Jalil et al. describe the beneficial impact of colonization with the bile acid 7 α -dehydroxylating bacterial species *Clostridium scindens* in the DSS model of inflammatory bowel disease. Colonization of gnotobiotic and normal mice was achieved, resulting in increased production of 7 α -dehydroxylated BA species and decreased primary BAs without changing total BA pool amounts. In both models, this colonization decreased pathology in the standard DSS model. Finally, analysis of legacy data from human IBD patients indicate that levels of 7 α -dehydroxy BAs are correlated with the presence of gene signatures associated with enterocytes, goblet cells and EECs, while increased primary BAs are associated with their pathologic loss. This provides at least a modest support for the human relevance of the observations in the DSS models.

Overall the results are relatively straightforward and the primary strength is the successful use of 2 different mouse models to test the impact of *C. scindens* colonization. The primary weakness is the exclusive use of the DSS model. Further work in more relevant models will be required to more firmly establish the relevance of the intriguing impact of increased production of secondary bile acids in IBD.

Referee #3 (Comments on Novelty/Model System for Author):

The approaches and models fit the purpose, studies seem to be well-performed. Interpretation is skewed towards own hypothesis without reference to existing literature using similar models. Medical impact limited so far - probiotic treatment of IBD will be complex and likely highly personalized

Referee #3 (Remarks for Author):

This is an interesting and timely manuscript addressing the role of microbiome-host interactions, particularly evaluating the role of secondary bile acids (DCA and LCA) formed by bacterial enzymatic reactions in the etiology of inflammatory bowel disease. Authors colonized the gut of gnotobiotic (Oligo-MM12) and conventional mice with *Clostridium scindens*, a bacterium known to exert 7 α -dehydroxylating activity on liver-derived primary bile acids, prior to induction of experimental colitis using DSS. It is shown that colonization with *C. scindens* indeed enhanced the presence of secondary bile acids in feces of mice, improved body weight gain after termination of DSS treatment and was associated with features of epithelial regeneration and specification, i.e., protection against colon injury. Furthermore, analysis of publicly available human datasets showed defective intestinal cell renewal and differentiation in colitis patients and positive correlation between levels of 7 α -dehydroxylated bile acids and expression of genes involved in these pathways. It is concluded that *C. scindens* administration could be promising for therapy in IBD. The paper is well-written and figures are clear. However, there are some issues that require attention and potentially adaptation of the manuscript.

General comments

1. During the past couple of years, a number of papers have been published in high-ranking journals on the role of (specific) secondary bile acids (LCA, isoLCA, 3oxoLCA) in protection against IBD, also in the context of DSS-induced experimental colitis, pointing towards a role in control of inflammation through activation of ROR γ , VDR and FXR. While the authors only focus on regeneration via TGR5, the potential contribution of anti-inflammatory actions should at least be mentioned and addressed. Paper would benefit from indications of inflammation, cytokines etc.
2. Introduction of a new bacterium into the gut ecosystem may induce all kinds of shifts in the existing bacterial populations: it may thus be that other protective metabolites are being formed after colonization with *C. scindens*. This should at least be mentioned. Paper would benefit from assessment of microbiome composition.
3. Authors use (relative) abundances of the various bile acid species in their correlation analyses. It appears that in the mouse experiments not only the abundance of secondary bile acids is increased but also those of the very hydrophilic muricholic acids (green colors in pie charts difficult to discern..). Since these bile acids are, in general, thought to be cell protective, an overall change in fecal bile acid composition may contribute to the phenotypic improvements. Might be an idea to calculate hydrophobicity index of fecal bile acids as a proxy. (see also point 2 of specific comments)

Specific comments

1. Would enhance readability if Oligo-MM12 mouse model was explained shortly.
2. What is the reason of more rapid weight gain in *C. scindens* treated mice after DSS treatment? Do they eat more? Since relative abundances of bile acids have been used: is the amount of feces produced by *C. scindens* treated mice larger than that of controls? Does this reflect in lower bile acid concentrations? If so, this may also contribute to the effects seen.
3. Tgr5 is also expressed in macrophages and its activation may therefore also impact on inflammatory processes (see also point 1 general comments)

EMM-2024-20494-V2**Bile acid 7 α -dehydroxylating bacteria accelerate injury-induced mucosal healing in the colon****Referee #1 (Remarks for Author):**

I read this manuscript with extreme interest. The authors present elegant data with state-of-the-art models that connects intestinal bacteria with bile acid metabolism, intestinal mucosa regeneration and inflammation. The data are well presented and the discussion is very well balanced. Nevertheless, few information are needed to complete the story.

We sincerely thank this reviewer for the positive feedback and thoughtful comments. We appreciate the recognition of our work and the opportunity to address the suggestions, which have allowed us to strengthen and further complete this study.

1. The authors should clearly present the bile acid profile both in the feces and in the serum or bile and discuss the modulation of the primary/secondary as well as the changes in the hydrophobicity index (which is a key point per se for prevention of intestinal damage)

We agree with the reviewer's suggestion that a more comprehensive BA profiling and analysis would strengthen our current study. To address this reviewer's comment, we have expanded our analysis of the feces and performed a detailed analysis of the plasma compartment. We did not collect bile due to the experimental design of our *in vivo* experiments in which we collect the tissues 2 hours following physiological feeding; see Methods and Protocols section. We now include BA quantification both as percentages and absolute amounts for the two mouse models, Oligo-MM¹² (Fig. EV1D, Tables EV1-2) and SPF-van (Fig. EV2E, Tables EV2F-2I). These additional representations confirm that the BA profiles are consistent between relative and absolute values in both fecal and plasma samples. Importantly, colonization with *C. scindens* in both Oligo-MM¹² and SPF-van mice does not alter the total plasma BA pool (Figs. EV1F, EV2I). However, it increases the proportion and amounts of 7 α -dehydroxylated BAs (Figs. EV1E, EV2G). Accordingly, colonization with *C. scindens* in both Oligo-MM¹² and SPF-van mice leads to a higher secondary-to-primary BA ratio both in feces (Figs. 1G, EV2H) and plasma (Figs. EV1F, EV2I).

As requested, we have also determined the hydrophobicity index. For this purpose, we used the BA data from the plasma compartment since calculating the fecal hydrophobicity index is challenging due to the lack of published hydrophobicity values for many fecal BAs, particularly unconjugated ones. We demonstrate that the selective changes in 7 α -dehydroxylated BA species do not affect the plasma hydrophobicity index (Figs. EV1G, EV2J). These additional data are now integrated and discussed in the revised manuscript (page 5, lines 95–114; page 7, lines 152–159).

2. Feces should eventually be studied for the 7-a-dehydroxilated activity (cell reports 2014; 7(1): 12.8). It is important to prove that in the aqueous phase of the feces there is a net 7a-DH activity on bile acids

As requested, we assessed 7 α -dehydroxylating activity in the feces of SPF-van mice colonized with *C. scindens*. To do this, we resuspended freshly collected fecal pellets in PBS before being incubated *in vitro* in BHI-S medium containing 100 μ M of CA at 37°C in an anaerobic chamber, and profiled CA and DCA 24 hours later. These data, showing significant conversion of CA into DCA, support the notion that the increased proportion of fecal DCA is caused by increased microbial 7 α -dehydroxylating activity. These novel findings are now integrated and discussed in the revised manuscript (Fig. 2D, pages 6-7, lines 138-142).

3. Gene expression profile is needed for both intestine and liver to prove if modulation of

bile acid species as well as hydrophobicity and quantity (pool) drives changes in the classical pathways (SHP, IBABP and FGF15 for the intestine and SHP, BSEP and CYP7A1 for the liver).

We thank this reviewer for this suggestion. To address this, we analyzed the hepatic and intestinal gene expression profiles of the above-mentioned FXR target genes. Our data, newly integrated into the revised manuscript, show no significant alterations in Oligo-MM¹² mice colonized with *C. scindens* (Figs. EV1H,I), suggesting that the observed BA changes do not influence these classical transcriptional mechanisms (page 5, lines 110-112).

Referee #2 (Comments on Novelty/Model System for Author):

DSS only is the primary limitation.

Referee #2 (Remarks for Author):

Jalil et al. describe the beneficial impact of colonization with the bile acid 7 α -dehydroxylating bacterial species *Clostridium scindens* in the DSS model of inflammatory bowel disease. Colonization of gnotobiotic and normal mice was achieved, resulting in increased production of 7 α -dehydroxylated BA species and decreased primary BAs without changing total BA pool amounts. In both models, this colonization decreased pathology in the standard DSS model. Finally, analysis of legacy data from human IBD patients indicate that levels of 7 α -dehydroxy BAs are correlated with the presence of gene signatures associated with enterocytes, goblet cells and EECs, while increased primary BAs are associated with their pathologic loss. This provides at least a modest support for the human relevance of the observations in the DSS models.

Overall the results are relatively straightforward and the primary strength is the successful use of 2 different mouse models to test the impact of *C. scindens* colonization. The primary weakness is the exclusive use of the DSS model. Further work in more relevant models will be required to more firmly establish the relevance of the intriguing impact of increased production of secondary bile acids in IBD.

We thank this reviewer for acknowledging the strength of our study in using two distinct mouse models to evaluate the impact of *C. scindens* colonization in DSS-induced experimental colitis. While we recognize that the exclusive use of the DSS model is a limitation, we opted to follow the editor's guidance, and not to pursue these additional experiments due to the challenges of obtaining approval from the animal authorities in Switzerland within the 3-month timespan for the revision. We plan to conduct follow-up studies in alternative models of IBD (e.g. *Il-10* knock-out mice, and adoptive transfer colitis model to assess the role of the *C. scindens*-TGR5 signaling axis in T cells), which will also be more appropriate for investigating the TGR5-mediated effect on the immune landscape. Given the relevance of this comment, we have explicitly acknowledged this limitation in the manuscript (end of discussion, page 12, lines 288-290).

Referee #3 (Comments on Novelty/Model System for Author):

The approaches and models fit the purpose, studies seem to be well-performed. Interpretation is skewed towards own hypothesis without reference to existing literature using similar models. Medical impact limited so far - probiotic treatment of IBD will be complex and likely highly personalized.

Referee #3 (Remarks for Author):

This is an interesting and timely manuscript addressing the role of microbiome-host interactions, particularly evaluating the role of secondary bile acids (DCA and LCA) formed by bacterial enzymatic reactions in the etiology of inflammatory bowel disease. Authors colonized the gut of gnotobiotic (Oligo-MM12) and conventional mice with *Clostridium scindens*, a bacterium known to exert 7 α -dehydroxylating activity on liver-derived primary bile acids, prior to induction of experimental colitis using DSS. It is shown that colonization with *C. scindens* indeed enhanced the presence of secondary bile acids in feces of mice, improved body weight gain after termination of DSS treatment and was associated with features of epithelial regeneration and specification, i.e., protection against colon injury. Furthermore, analysis of publicly available human datasets showed defective intestinal cell renewal and differentiation in colitis patients and positive correlation between levels of 7 α -dehydroxylated bile acids and expression of genes involved in these pathways. It is concluded that *C. scindens* administration could be promising for therapy in IBD. The paper is well-written and figures are clear. However, there are some issues that require attention and potentially adaptation of the manuscript.

We thank this reviewer for the positive feedback, and for finding our manuscript timely and well-written. We also appreciate the constructive suggestions, which we have carefully addressed and incorporated in the revised version, as outlined below.

General comments

1. During the past couple of years, a number of papers have been published in high-ranking journals on the role of (specific) secondary bile acids (LCA, isoLCA, 3oxoLCA) in protection against IBD, also in the context of DSS-induced experimental colitis, pointing towards a role in control of inflammation through activation of ROR γ , VDR and FXR. While the authors only focus on regeneration via TGR5, the potential contribution of anti-inflammatory actions should at least be mentioned and addressed. Paper would benefit from indications of inflammation, cytokines etc.

We thank the reviewer for this insightful comment, which indeed highlights an essential aspect of BA signaling in IBD. Although the main center of attention of this study is focused on the preclinical investigation of *C. scindens* as a live biotherapeutic product stimulating mucosal regeneration, we recognize the relevance of discussing the established anti-inflammatory roles of secondary BAs. To address this, we have added a dedicated paragraph to the discussion, elaborating on the role of secondary BA signaling in controlling inflammation through pathways such as ROR γ , VDR and FXR (page 12, lines 272-285). Additionally, we have extended our analysis to include local and systemic indicators of inflammation after DSS-induced acute experimental colitis. More specifically, we profiled the local immune landscape in the colon by using flow cytometry (Fig. R1A). Consistent with the immunosuppressive role of TGR5 in the colon (PMID: 22046243), genetic loss of TGR5 in DSS-treated mice was associated with an increase in leukocytes, pro-inflammatory monocytes (Ly6C^{hi}), macrophages, and dendritic cells (Fig. R1B). In SPF-Van TGR5 wild-type mice, colonization of *C. scindens*, was less effective, and only elicited a trend toward reducing the colonic inflammatory response (Fig. R1B). In addition, our cytokine analysis in plasma showed no significant changes, with cytokine concentration remaining very low (Fig. R1C), indicating an absence or negligible systemic inflammatory response.

Figure R1

Figure R1. (A and B) 10-week-old male SPF-Van wild-type (*Tgr5*^{+/+}) and TGR5 knock-out (*Tgr5*^{-/-}) mice were gavaged daily for 5 days with live *C. scindens* (10^8 CFU - *Tgr5*^{+/+} SPF-Van + *C. scindens* ($n = 5$) and *Tgr5*^{-/-} SPF-Van + *C. scindens* ($n = 6$)) or vehicle (PBS - *Tgr5*^{+/+} SPF-Van ($n = 6$) and *Tgr5*^{-/-} SPF-Van ($n = 5$)). Experimental colitis was induced by a 7-day treatment with DSS (2.5% in drinking water). Flow cytometry gating strategy (A) and quantification (B) of the proportion of leukocytes (CD45⁺), monocytes (CD45⁺, Ly6C⁺, CD11b⁺), neutrophils (CD45⁺, Ly6G⁺, CD11b⁺), macrophages (CD45⁺, F4/80⁺, CD11b⁺) and dendritic cells (CD45⁺, CD11c⁺, MHCII⁺) isolated for colon of mice described above. Proportion of immune cells (% of live cells) isolated from colon is represented. (C) Flow cytometry analysis of the indicated cytokines in plasma of mice described in A at the end of the DSS treatment. Graphs represent mean \pm SEM. n refers to biological replicates. P values (exact values) were calculated using 1-way ANOVA followed by Bonferroni's post hoc correction (A,B).

These observations suggest that while the DSS model used in our study is well-suited for studying local tissue injury and repair mechanisms, it has limitations for exploring broader immunomodulatory effects. Indeed, it has been shown that the absence of a gut microbiome

(e.g., germ-free mouse models) or its perturbation through antibiotic treatment can attenuate DSS-induced colitis severity, including inflammatory response (PMIDs: 22665990, 10750645). Therefore, future studies using more immune-centered colitis models or chronic colitis models might better elucidate the potential synergy between TGR5 activation and *C. scindens* in regulating immune responses. For this reason, we opted not to add these new data in the revised manuscript. Nonetheless, we are willing to incorporate these new data into the manuscript if the reviewers consider that it would provide relevant insights and strengthen the study.

2. Introduction of a new bacterium into the gut ecosystem may induce all kinds of shifts in the existing bacterial populations: it may thus be that other protective metabolites are being formed after colonization with *C. scindens*. This should at least be mentioned. Paper would benefit from assessment of microbiome composition.

We agree with the reviewer that colonization with *C. scindens* may lead to the production of additional protective metabolites. However, as all the observed effects depend on the BA receptor TGR5, our findings support the notion that the 7α -dehydroxylated BAs are prime mediators of these effects. To comply with the reviewer's comment, we have mentioned this point in the revised version of the manuscript (pages 10-11, lines 237-242).

In the previous version of this manuscript, we already included 16S rRNA sequencing analysis (Fig. EV2) to evaluate the extent of dysbiosis triggered by vancomycin preconditioning and/or *C. scindens* colonization. As expected, these data revealed a significant modulation of the gut microbiome ecosystem induced by vancomycin (Fig. EV2D), but no further modification of the overall family composition by *C. scindens* (Figs. EV2A,D). Although out of the scope of the current study, additional in-depth investigations using metagenomics and metabolomics could be envisioned to gain new insights into how microbial species/metabolites, other than 7α -dehydroxylated BAs, may further contribute to the observed phenotype.

3. Authors use (relative) abundances of the various bile acid species in their correlation analyses. It appears that in the mouse experiments not only the abundance of secondary bile acids is increased but also those of the very hydrophilic muricholic acids (green colors in pie charts difficult to discern..). Since these bile acids are, in general, thought to be cell protective, an overall change in fecal bile acid composition may contribute to the phenotypic improvements. Might be an idea to calculate hydrophobicity index of fecal bile acids as a proxy.(see also point 2 of specific comments).

We thank the reviewer for this insightful comment. To improve data representation, we have replaced the colors of the pie charts, allowing for more precise visualization of BA composition. In addition, the BA data are now shown as both percentages and absolute amounts (Figs. 1D-G, 2E, EV1C-F, EV2E-I, Tables EV1-4).

Regarding the muricholic acids (MCAs), we observed slight modulations in their fecal abundances in both OligoMM¹² and SPF-van mice colonized with *C. scindens* (Fig. R2). However, the patterns of change differed between the two mouse models. In Oligo-MM¹² mice colonized with *C. scindens*, α MCA levels decreased, while ω MCA levels increased (Fig. R2A). Conversely, in SPF-van mice colonized with *C. scindens*, β MCA levels decreased while TaMCA, and T β MCA levels increased (Fig. R2B). These differences in MCA modulation between the two mouse models indicate that the tissue regeneration and mucosal healing phenotypes observed in both models, driven by the *C. scindens*-TGR5 axis, are unlikely to be explained by changes in MCA species. Additionally, it is important to note that MCAs are weak agonist of TGR5 (EC₅₀=4.89 μ M; PMID:18307294).

Assessment of the fecal hydrophobicity index was challenging due to the lack of published hydrophobicity values for many fecal BAs, particularly unconjugated ones. However, we calculated the hydrophobicity index of plasma BAs in both Oligo-MM¹² (Fig. EV1G) and SPF-van mice (Fig. EV2J). The selective changes in BA species following *C. scindens* colonization did not

affect the plasma hydrophobicity index. These additional data are now integrated and discussed in the revised manuscript (page 5, lines 95–114; page 7, lines 152–159).

Figure R2

Figure R2. (A) Fecal BA of Oligo-MM¹² mice gavaged with live *C. scindens* (10^7 CFU – Oligo-MM¹² + *C. scindens*) or PBS (control – Oligo-MM¹²) 10 days after the experiment start ($n = 7$ /group). (B) Fecal BA of SPF-Van mice gavaged daily for 5 days with live *C. scindens* (10^8 CFU – SPF-Van + *C. scindens*) ($n = 11$) or PBS (control – SPF-Van) ($n = 10$). Graphs represent mean \pm SEM. n refers to biological replicates. P values (exact values) were calculated using 2-tailed Student's t-test (A,B).

Specific comments

1. Would enhance readability if Oligo-MM12 mouse model was explained shortly.

As suggested, we have added (page 5, lines 92-95) a brief explanation of the Oligo-MM¹² mouse model in the Introduction to enhance readability and provide the necessary context for the readers.

2. What is the reason of more rapid weight gain in *C. scindens* treated mice after DSS treatment? Do they eat more? Since relative abundances of bile acids have been used: is the amount of feces produced by *C. scindens* treated mice larger than that of controls? Does this reflect in lower bile acid concentrations? If so, this may also contribute to the effects seen.

We thank the reviewer for these pertinent comments. Here below our replies:

1/ Do *C. scindens* treated mice eat more? This comment on food intake is relevant in view of previous reports (PMID: 34031591; 32699194) identifying BAs as postprandial regulators of eating behavior in the hypothalamus and vagal afferent neurons. Monitoring body weight loss and progressive gain in DSS-induced colitis is the standard phenotyping procedure to assess injury severity and healing capacity following the insult, but in this particular case, it could also reflect changes in food intake. We initially did not measure food intake because DSS-treated mice typically experience severe illness and reduced appetite during the experimental period. However, to address this point, we assessed food intake with Promethion metabolic cages (Sable Systems) in SPF-van mice fed a chow diet. In this cohort, *C. scindens* administration did not result in any significant changes in food intake (Fig. R3), suggesting that the overall impact of *C. scindens* colonization on BA composition remodeling does not influence central control of energy homeostasis. Therefore, these data indicate that the changes in BA composition but not

pool size by *C. scindens* are insufficient to modulate central actions, such as food intake behavior.

Figure R3

Figure R3. Food intake measurement over 24h period of chow diet-fed male SPF-Van mice gavaged daily for 5 days with live *C. scindens* (10^8 CFU – SPF-Van + *C. scindens*) ($n = 12$) or PBS (control – SPF-Van) ($n = 10$). Graphs represent mean \pm SEM. n refers to biological replicates. P values were calculated using 2-tailed Student's t-test.

2/ is the amount of feces produced by *C. scindens* treated mice larger than that of controls? Although we could not collect the fecal pellets post-DSS, we measured fecal weight of control and *C. scindens* cohorts collected over a period of 24 hours in a parallel high-fat diet study.

As illustrated in the figure below, *C. scindens* administration did not affect the feces mass (Fig. R4).

Figure R4

Figure R4. Feces measurement over 24h period of high-fat diet-fed male SPF-Van mice gavaged daily for 5 days with live *C. scindens* (10^8 CFU – SPF-Van + *C. scindens*) ($n = 7$) or PBS (control – SPF-Van) ($n = 7$). Graphs represent mean \pm SEM. n refers to biological replicates. P values were calculated using 2-tailed Student's t-test.

3/ Does this reflect in lower bile acid concentrations? DSS treatment induces diarrhea, compromising the collection of fecal pellets for accurate weight measurements. Therefore, we measured the plasma BA SPF-van mice after 7 days of DSS treatment followed by a 3-day recovery phase. Although individual plasma BA levels varied considerably within each group (most likely because some mice, still recovering from the DSS challenge, did not eat (mice were euthanized after 2 hours of refeeding with free access to food)), colonization with *C. scindens* resulted in the expected BA remodeling with blunted plasma TCA and increased DCA levels (Fig. R5). In contrast, administration of *C. scindens* did not affect the total BA pool size in the plasma compartment (Fig. R5).

Figure R5

Figure R5. Plasma BA concentration of SPF-Van mice gavaged daily for 5 days with live *C. scindens* (10^8 CFU – SPF-Van + *C. scindens* ($n = 10$)) or vehicle (PBS - SPF-Van ($n = 11$)). Experimental colitis was induced by a 7-day treatment with DSS (2.5% in drinking water) followed by 3 days of drinking water (recovery period). Graphs represent mean \pm SEM. n refers to biological replicates. P values were calculated using 2-tailed Student's t-test.

Altogether, the absence of changes in food intake, fecal mass and total plasma BA concentration further supports our conclusions that the more rapid weight gain observed in *C. scindens*-colonized mice after DSS treatment is due to improved intestinal regeneration and mucosal healing.

3. *Tgr5* is also expressed in macrophages and its activation may therefore also impact ion inflammatory processes (see also point 1 general comments)

In addition to the leukocyte phenotyping (Fig. R1), we have analyzed the proportion of pro-inflammatory M1 macrophages (CD11b⁺, F4/80⁺, CD86⁺) within the colonic tissue after DSS-induced acute colitis by flow cytometry. Interestingly, *Tgr5*^{+/+} SPF-Van mice showed a reduction in M1 macrophages compared to *Tgr5*^{-/-} SPF-Van mice (Fig. R6), further supporting the role of TGR5 in macrophage polarization. Although amendment with *C. scindens* in *Tgr5*^{+/+} SPF-Van mice showed a trend toward reducing the colonic inflammatory response, colonization of *Tgr5*^{+/+} SPF-Van mice with *C. scindens* did not lead to a significant reduction in M1 macrophages. As discussed above, these observations suggest that while the DSS model used in our study is well-suited for studying local tissue injury and repair mechanisms, it has limitations for exploring broader immunomodulatory effects. Indeed, it has been shown that the absence of a gut microbiome (e.g., germ-free mouse models) or its perturbation through antibiotic treatment can attenuate DSS-induced colitis severity, including inflammatory response (PMIDs: 22665990, 10750645).

Figure R6

Figure R6. Proportion of M1 macrophages in colon of 10-week-old male SPF-Van TGR5 wild-type (*Tgr5*^{+/+}) and TGR5 knock-out (*Tgr5*^{-/-}) mice gavaged daily for 5 days with live *C. scindens* (10^8 CFU – *Tgr5*^{+/+} SPF-Van + *C. scindens* ($n=5$) and *Tgr5*^{-/-} SPF-Van + *C. scindens* ($n=6$)) or vehicle (PBS - *Tgr5*^{+/+} SPF-Van ($n = 6$) and *Tgr5*^{-/-} SPF-Van ($n = 5$)). Graphs represent mean \pm SEM. n refers to biological replicates. P values (exact values) were calculated using 1-way ANOVA followed by Bonferroni's post hoc correction.

30th Jan 2025

Dear Prof. Schoonjans,

Thank you for submitting your revised manuscript to EMBO Molecular Medicine. We have now received the enclosed report from the two referees who re-assessed your work. As you will see, the referees are now supportive, and I am pleased to inform you that we will be able to accept your manuscript pending the following amendments:

1. Data availability : please remove the reviewer access link and reviewer code and make sure the datasets are made publicly available upon the acceptance of the manuscript.
2. Every published paper now includes a 'Synopsis' to further enhance discoverability. Synopses are displayed on the journal webpage and are freely accessible to all readers. They include a short stand first (maximum of 300 characters, including space) as well as 2-5 one-sentences bullet points that summarizes the paper. Please write the bullet points to summarize the key NEW findings. They should be designed to be complementary to the abstract - i.e. not repeat the same text. We encourage inclusion of key acronyms and quantitative information (maximum of 30 words / bullet point). Please use the passive voice. Please attach these in a separate file or send them by email, we will incorporate them accordingly.
3. The synopsis image is too large. Please provide an updated version as a PNG file 550 px wide x 300-600 px high.
4. Please address the following issues in figure legends:
 - Please note that the exact p values are not provided in the legends of figures 1C, E, F, G, I; 2C, G, N; 3B, D, F, J, K, L; 4e, EV1B, EV2 A, F, H, L; EV3 B.
 - Please note that the exact p values are not provided in the legends of figures 1C, E, F, G, I; 2C, G, N; 3B, D, F, J, K, L; 4e, EV1B, EV2 A, F, H, L; EV3 B.

Please submit your revised manuscript within two weeks. I look forward to seeing a revised form of your manuscript as soon as possible.

Kind regards,
Jingyi

Jingyi Hou
Senior Editor
EMBO Molecular Medicine

*** Instructions to submit your revised manuscript ***

- 1) a .docx formatted version of the manuscript text (including Figure legends and tables)
- 2) Separate figure files*
- 3) supplemental information as Expanded View and/or Appendix. Please carefully check the authors guidelines for formatting

Expanded view and Appendix figures and tables at
<https://www.embopress.org/page/journal/17574684/authorguide#expandedview>

4) a letter INCLUDING the reviewer's reports and your detailed responses to their comments (as Word file).

5) The paper explained: EMBO Molecular Medicine articles are accompanied by a summary of the articles to emphasize the major findings in the paper and their medical implications for the non-specialist reader. Please provide a draft summary of your article highlighting

6) Author contributions: the contribution of every author must be detailed in a separate section.

7) EMBO Molecular Medicine now requires a complete author checklist (<https://www.embopress.org/page/journal/17574684/authorguide>) to be submitted with all revised manuscripts. Please use the checklist as guideline for the sort of information we need WITHIN the manuscript. The checklist should only be filled with page numbers where the information can be found. This is particularly important for animal reporting, antibody dilutions (missing) and exact values and n that should be indicated instead of a range.

8) Every published paper now includes a 'Synopsis' to further enhance discoverability. Synopses are displayed on the journal webpage and are freely accessible to all readers. They include a short stand first (maximum of 300 characters, including space) as well as 2-5 one sentence bullet points that summarise the paper. Please write the bullet points to summarise the key NEW findings. They should be designed to be complementary to the abstract - i.e. not repeat the same text. We encourage inclusion of key acronyms and quantitative information (maximum of 30 words / bullet point). Please use the passive voice. Please attach these in a separate file or send them by email, we will incorporate them accordingly.

You are also welcome to suggest a striking image or visual abstract to illustrate your article. If you do please provide a jpeg file 550 px-wide x 300-600px high.

9) A Conflict of Interest statement should be provided in the main text

10) Please note that we now mandate that all corresponding authors list an ORCID digital identifier. This takes <90 seconds to complete. We encourage all authors to supply an ORCID identifier, which will be linked to their name for unambiguous name identification.

Currently, our records indicate that the ORCID for your account is 0000-0003-1247-4265.

Link Not Available

11) Include a Reagents and Tools Table as part of the Methods section, which can be downloaded from our author guidelines (<https://www.embopress.org/page/journal/17574684/authorguide#structuredmethods>)

Photos 400-800 DPI

*Additional important information regarding figures and illustrations can be found at
<https://bit.ly/EMBOPressFigurePreparationGuideline>. See also figure legend preparation guidelines:
<https://www.embopress.org/page/journal/17574684/authorguide#figureformat>

***** Reviewer's comments *****

Referee #1 (Comments on Novelty/Model System for Author):

The authors answered all the questions and the new data strongly support their original hypothesis and conclusions.

Referee #1 (Remarks for Author):

The authors answered all the questions and the new data strongly support their original hypothesis and conclusions.

Referee #3 (Comments on Novelty/Model System for Author):

N/A

Referee #3 (Remarks for Author):

Authors did a great job in improving the manuscript, no further remarks

All editorial and formatting issues were resolved by the authors.

5th Feb 2025

Dear Prof. Schoonjans,

I am pleased to inform you that your manuscript is accepted for publication and is now being sent to our publisher to be included in the next available issue of EMBO Molecular Medicine. It's been a pleasure working with you to bring this paper to acceptance stage.

Kind regards,
Jingyi

Jingyi Hou
Senior Editor
EMBO Molecular Medicine
